



# OceanVar2.0: an open-source variational ocean data assimilation scheme. Sensitivity to altimetry sea level anomaly assimilation

Paolo Oddo[1,2], Mario Adani[2], Francesco Carere[2], Andrea Cipollone[2], Anna Chiara Goglio[2], Eric Jansen[2],
Ali Aydogdu[2], Francesca Mele[2], Italo Epicoco[2], Jenny Pistoia[2], Emanuela Clementi[2], Nadia Pinardi[1,2],
Simona Masina[2]

[1]Bologna University, Department of Physics and Astronomy, Italy.
[2]CMCC Foundation - Euro-Mediterranean Center on Climate Change, Italy.

*Correspondence to*: Paolo Oddo (paolo.oddo@unibo.it)

**Abstract**

This study presents recent developments of the OceanVar oceanographic three-dimensional variational data assimilation scheme to create OceanVar2.0. The code has been extensively revised to integrate past developments into a single, consistent,
fully parallelized framework. In OceanVar, the background error covariance matrix is decomposed into a sequence of physically based linear operators, allowing for individual analysis of specific error matrix components. We focus on the sea level  operator, which provides correlation between Sea Level Anomaly, temperature and salinity  increments. OceanVar2.0 offers the flexibility to use either a dynamic height or a barotropic model for closed domains as sea level operators. A diffusive operator to model the horizontal error correlations, replacing the previously used recursive filter, has been implemented. The
new code was tested in the Mediterranean Sea and the quality of the analysis assessed by comparing background estimates with observations for the period January-December 2021. The results highlight the better skill of the barotropic model operator with respect to the dynamic height one due to the assumptions required for the level-of-no-motion. Furthermore, we present a method to assimilate along track satellite altimetry considering a forecasting model with tides.

## 1. Introduction

Understanding the past state of the ocean and predicting its future behaviour is critical for sustainable development of human activities and to find a solution to climate change mitigation and adaptation strategies. Oceans are a key component of the earth climate system, and they require specific data assimilation schemes due to the sparsity of data in the ocean interior. However, satellites help to cover almost synoptically the ocean surface and satellite altimetry contains information of the subsurface thermohaline structure that is key to obtain best estimates of the ocean variability at depth. Satellite altimetric sea



level data are available from 1992, and altimeters have increased in coverage in the past ten years. Satellite altimetry and satellite-derived sea surface temperature are crucial in bridging gaps left by in-situ observations. However, their effective integration into model corrections requires advanced extrapolation algorithms, as demonstrated in the influential work of De Mey and Robinson (1987).

There are different methodologies for ocean data assimilation, each with its own strengths and weaknesses. Among the inverse

problem theory, the two most used approaches are the variational and the Kalman filter (Carrassi *et al.*, 2018). Schemes based on Monte-Carlo algorithms, such as the Particle filter, have been proven to be successful on low-dimensional systems and become feasible for high-dimensional geophysical systems only recently (Van Leeuwen *et al.*, 2019). The choice of data assimilation method depends on factors such as the type of data available, the desired forecast horizon, and the computational resources.

Recent machine learning (ML) advancements offer potential optimizations for ocean data assimilation (e.g., Barthélémy *et al.*, 2022; Beauchamp *et al.,* 2023). ML can refine errors representation and reveal complex relationships, improving accuracy. To fully leverage ML and new data streams, modular and flexible data assimilation codes are essential. As research progresses, these advancements will significantly enhance our ability to understand and predict ocean behaviour.

The data assimilation community has made substantial strides in developing shared software tools like PDAF (Nerger *et al.*,

2005), ROMS-4DVAR (Moore *et al.*, 2011), DART (Data Assimilation Research Testbed, Anderson *et al.*, 2009), just to mention a few. Each of these tools offers unique capabilities and has contributed significantly to the field. However, the increasing complexity of data assimilation problems, particularly with the adoption of ML, demands continued research and development of new tools.

OceanVar is the data assimilation scheme presented and discussed in this study. It was first introduced by Dobricic and Pinardi

(2008, hereafter DP08) and it is based on a three-dimensional variational method. OceanVar features a modular design that allows for flexibility in incorporating diverse data sources and error covariance representations. This adaptability has made it suitable for a wide range of applications and research needs. The scheme has been extensively used in several operational and reanalysis systems, as evidenced by numerous publications (Dobricic *et al.*, 2007, Storto *et al.*, 2016, Escudier *et al.*, 2021, Ciliberti *et al.*, 2022, Coppini *et al.,* 2023). The code has been also used to develop and test new hybrid variational formulations

(Oddo *et al.* 2016, Storto *et al.*, 2018); it has been interfaced with Artificial Intelligence based observational operators (Storto *et al.*, 2021), and has been used to test and develop numerous new schemes and features (Nilsson *et al.*, 2012; Dobricic *et al.*, 2015; Aydogdu *et al.*, 2016; Storto *et al.*, 2020; Storto and Oddo 2019, Teruzzi *et al.*, 2014, 2018).

The pervasive use of the code in diverse applications has led to a proliferation of versions that are not always consistent or compatible with each other, due to different requirements and operational contexts. The development of OceanVar2.0

(hereafter OceanVar2 for improved readability) involved comprehensive testing and debugging of the underlying OceanVar framework, which successfully ensured the consistency and reproducibility of results generated by its various computational modules. This rigorous development positions OceanVar2.0 as what we consider a leading advancement in ocean data assimilation specifically for satellite altimetry sea level anomalies



In OceanVar2, we included a new operator to model the horizontal error correlations. In DP08 and all subsequent
developments, horizontal covariance was approximated using the recursive filter (Lorenc, 1992; Hayden and Purser, 1995).
The filter is conceptually simple, typically requiring only a few iterations to approximate the Gaussian function, and its
application on a horizontal grid can be split into two independent directions (Purser *et al.*, 2003). However, in cases of spatially
or temporally varying correlation radii, the computational advantage of the recursive filter may be questioned (Purser *et al.*,
2023). To give the code more flexibility in terms of horizontal correlation radii while ensuring computational efficiency, we
modelled the horizontal correlation by a repeated application of the Laplacian operator, which is also the solution of the
horizontal diffusion equation (e.g., Derber and Rosati, 1989; Weaver and Courtier, 2001).

In this paper we want to show the capability of OceanVar2 to effectively assimilate along track satellite altimetry with the use
of a barotropic model operator and multivariate sea level, temperature and salinity statistics. We believe this is still an open
question in the data assimilation community and we offer a revisited formulation of the barotropic model defined by DP08 and
for the first time a detailed comparison with the dynamic height operator. Additionally, OceanVar2 is applied to a newly
developed Mediterranean Sea circulation forecasting model (Clementi *et al.*, 2023) that considers tidal forcing. Tides are
becoming an essential component of the resolved variability of the ocean general circulation, and they cannot anymore be
neglected in numerical ocean circulation models (Arbic, 2022). Satellite altimeters sample tides along their track as well as the
mesoscales. Using OceanVar2 we present a preliminary solution to the problem of assimilation in presence of tidal components
both in model and observations.

The manuscript is organized as follows. After the introduction, Section 1 provides a general overview of the variational
formulation and the characteristics of the OceanVar2. Section 3 presents the background and the observational error covariance
matrixes formulation and their specific operators. In Section 4 we describe the experimental set-up and the altimetry
assimilation methods. In Section 5 we discuss the results. In Section 6 an overview on the code performances is provided.
Finally, Section 7 provides the summary and conclusions.

## 2. Formulation of the Variational Assimilation Scheme

The general cost function in three-dimensional variational data assimilation is defined as:

$$J(x) = \frac{1}{2}(x - x_b)^T B^{-1}(x - x_b) + \frac{1}{2}(y - h[x])^T R^{-1}(y - h[x]) \tag{1}$$

where, $x$ is the true state vector for the ocean model prognostic variables, $x_b$ is the background state vector, $B$ is the background
error covariance matrix, $h$ is the observational operator, $y$ are the observations, $R$ is the observational error covariance matrix
and $T$ indicates the matrix transpose. The state vector contains the following model state variables:

$$x = [T, S, u, v, \eta]^T$$

where $T$ is the three-dimensional temperature field, $S$ the three-dimensional salinity field, $u$ and $v$ are the total horizontal
velocity components and $\eta$ the two-dimensional sea surface height.



The increments are defined as:

$$\delta \boldsymbol{x} = \boldsymbol{x} - \boldsymbol{x}_b.$$

When the observations are dynamical variables of the numerical model, the observational operator is a spatial interpolation algorithm to map the model solution into the observation locations. When the observed variables are indirectly related to model

dynamical variables, the observational operator $\boldsymbol{h}$ may be nonlinear. Using a Taylor expansion of the observational operator around the background, we can write:

$$\boldsymbol{h}[\boldsymbol{x} + \boldsymbol{x}_b - \boldsymbol{x}_b] = \boldsymbol{h}[\boldsymbol{x}_b + \delta \boldsymbol{x}] \approx \boldsymbol{h}[\boldsymbol{x}_b] + \mathbf{H}\delta \boldsymbol{x}$$

where $\mathbf{H}$ is the Jacobian matrix of $\boldsymbol{h}$ at $\boldsymbol{x} = \boldsymbol{x}_b$:

$$\mathbf{H} = \frac{\partial h}{\partial x}\Big|_{x=x_b}$$

then:

$$\boldsymbol{y} - \boldsymbol{h}[\boldsymbol{x}] = \boldsymbol{y} - \boldsymbol{h}[\boldsymbol{x}_b + \delta \boldsymbol{x}] \approx \{\boldsymbol{y} - \boldsymbol{h}[\boldsymbol{x}_b]\} - \mathbf{H}\delta \boldsymbol{x} \tag{2}$$

In case of linear $\boldsymbol{h}$, (2) is exact.

Defining the misfit (or innovation), $\boldsymbol{d} = \boldsymbol{y} - \boldsymbol{h}[\boldsymbol{x}_b]$, we can rewrite (1) linearized around the background state (*e.g.*, Lorenc, 1997) as:

$$J(\delta \boldsymbol{x}) = \frac{1}{2}\delta \boldsymbol{x}^T \mathbf{B}^{-1} \delta \boldsymbol{x} + \frac{1}{2}(\boldsymbol{d} - \mathbf{H}\delta \boldsymbol{x})^T \mathbf{R}^{-1}(\boldsymbol{d} - \mathbf{H}\delta \boldsymbol{x}) \tag{3}$$

The misfit in OceanVar is estimated using the FGAT (First Guess at Appropriate Time) method.

The minimum of $J(\delta \boldsymbol{x})$ in (3) is obtained for $\boldsymbol{x}_a = \boldsymbol{x}$, so that we obtain the increments $\delta \boldsymbol{x}$ that give the analysis at an instantaneous time:

$$\boldsymbol{x}_a = \boldsymbol{x}_b + \delta x \approx \boldsymbol{x}$$

The valid time of the increment using the FGAT algorithm has been discussed and investigated in literature (see Massart *et al.*, 2010). If the assimilation cycle goes from $t_n$ to $t_{n+1}$. the misfits are computed using the FGAT in the window [$t_n$; $t_{n+1}$]:

$$\boldsymbol{d}|_{[t_n;t_{n+1}]} = (\boldsymbol{y} - \boldsymbol{h}[\boldsymbol{x}_b])|_{[t_n;t_{n+1}]}$$

and the increments are applied at $t_{n+1}$. Formally our analysis is defined as the instantaneous field:

$$\boldsymbol{x}_a(t_{n+1}) = \boldsymbol{x}_b(t_{n+1}) + \delta x$$

where $\boldsymbol{x}_b(t_{n+1})$ is the instantaneous background field simulated by the nonlinear ocean model starting from $t_n$.

Existence and uniqueness of $\boldsymbol{x}_a$ is guaranteed because $J$ is quadratic with $\mathbf{R}$ and $\mathbf{B}$ positive defined and can be found by forcing the gradient of the cost function to zero. The gradient of (3) is:

$$\nabla J(\delta \boldsymbol{x}) = (\mathbf{B}^{-1} + \mathbf{H}^T \mathbf{R}^{-1} \mathbf{H})\delta \boldsymbol{x} - \mathbf{H}^T \mathbf{R}^{-1}\boldsymbol{d} \tag{4}$$

Following DP08, the OceanVar scheme assumes that the $\mathbf{B}$ matrix can be rewritten and decomposed as:

$$\mathbf{B} = \mathbf{V}\mathbf{V}^T \tag{5}$$

and the cost function may equivalently be minimized using a new control variable $\boldsymbol{v}$ (*e.g.*, Lorenc, 1997) defined using the transformation matrix $\mathbf{V}^+$:





$$v = \mathbf{V}^+ \delta \mathbf{x} \tag{6}$$

Where the superscript "+" indicates the generalized inverse. The vector $v$ is defined on the control space, and the vector $\delta \mathbf{x}$ on
the physical space. The cost function (3) now has the form:

$$J = \frac{1}{2} v^T v + \frac{1}{2} (\mathbf{H} V v - d)^T R^{-1} (\mathbf{H} V v - d) \tag{7}$$

### 3. Modelling the background error covariance matrix

The transformation matrix $\mathbf{V}$ is modelled at each minimization iteration as a sequence of linear operators (*e.g.*, Weaver *et al.*, 2003). In this way $\mathbf{V}$ successively transforms increments in the control space towards final increments in the physical space.
In OceanVar and OceanVar2 the matrix $\mathbf{V}$ is defined in the following way:

$$\mathbf{V} = \mathbf{V}_D \mathbf{V}_{u,v} \mathbf{V}_\eta \mathbf{V}_H \mathbf{V}_V. \tag{8}$$

From right to left $\mathbf{V}_v$ defines the vertical error covariance, $\mathbf{V}_H$ the horizontal error correlation, $\mathbf{V}_\eta$ is the sea level operator containing correlation between temperature, salinity and sea surface elevation, and $\mathbf{V}_{u,v}$ forces a geostrophic balance between temperature, salinity and the velocity components. Finally, $\mathbf{V}_D$ is a divergence-damping operator avoiding spurious currents
close to the coast in the presence of complex coastlines, as defined by DB08.

The vertical transformation operator $\mathbf{V}_v$ has the form:

$$\mathbf{V}_V = \mathbf{S_c} \mathbf{\Lambda_c}^{1/2} \tag{9}$$

where columns of $\mathbf{S_c}$ contain eigenvectors and $\mathbf{\Lambda_c}$ is a diagonal matrix with eigenvalues of multivariate Empirical Orthogonal Functions (EOFs). In OceanVar code, the EOFs can be defined pointwise (Coppini *et al.*, 2023) or by regions (DP08).
To account for horizontal field correlations, $\mathbf{V}_H$ is considered as the discretized form of the diffusive operator:

$$\mathbf{V}_H = \boldsymbol{\nabla_H} \cdot (k_c \boldsymbol{\nabla_H} C)$$

where $\boldsymbol{\nabla_H}$ is the horizontal differential operator, $k_c$ is the spatially variable diffusivity coefficient corresponding to horizontal correlation lengths, and $C$ is a generic increment. The operator is discretized with a Euler-Backward implicit scheme solved by means of LU decomposition of a tri-diagonal matrix, following Hoffman *et al.* (2011). Assuming a gaussian solution
(Weaver and Courtier 2001), the relation between $k_c$ and the horizontal correlation radius is:

$$R_H = \sqrt{2 k_c \Delta t}$$

Where $R_H$, in meters, is the horizontal correlation radius, and $\Delta t = 1$ second is the pseudo time-step used to integrate the diffusion equation.

The present formulation of the OceanVar2 allows the possibility to compute correction also for the velocity field. The
$\mathbf{V}_{u,v}$ operator calculates the velocity corrections assuming geostrophic balance. The advantage of the geostrophic assumption is that it requires only a small computational effort, but the disadvantage is that it is not valid at the equator and may produce velocity vectors orthogonal to the coast. Enforcing the zero-boundary condition for the velocity component perpendicular to the coast, the divergence component of the velocity field may become unrealistically large. Therefore, the divergence damping



operator $\mathbf{V}_D$ in (8) is implemented to damp velocity divergence near coasts, while maintaining the vorticity part unchanged.

Details on the implementation of the divergence damping operator are provided in DP08. As highlighted in DB08, the sequence of operator multiplication is critical and determined through a combination of physical reasoning and iterative experimentation. Initially, all increments are projected onto multivariate EOFs for sea level, temperature, and salinity, as these effectively capture ocean stratification (Sanchez de la Lama *et al.*, 2016) and the relationship with sea level for assimilation purposes (De Mey and Robinson, 1987). Next, the increments are distributed horizontally. Following this, adjustments due to the sea level

operator are computed based on the vertically projected temperature and salinity increments. Subsequently, increments in horizontal velocities are derived, and the process concludes with the application of a divergence damping filter.

### 3.1 The sea level and the velocity operators

Dobricic *et al.* (2007) found that the vertical EOFs computed from the covariance between temperature, salinity and sea level

could produce corrections that are not geostrophically balanced and proved that the enforcement of the geostrophic relationship for the sea level in the error covariance matrix has a significant positive impact on the accuracy of the analyses. Thus, in OceanVar2 the sea surface height increments from EOFs projections are overwritten using a sea level operator. Two different sea level operators, with different levels of complexity, are implemented in OceanVar2.

The first is the commonly used Dynamic Height operator $\left(\mathbf{V}_\eta = \mathbf{V}_{DH}\right)$, which we write here as:

$$\mathbf{V}_{\text{DH}} = \delta\eta^{DH} = -\frac{1}{\rho_0}\int_{-D}^{0}[-\alpha\delta T + \beta\delta S]\, dz \tag{10}$$

Where $D$ is a uniform reference level, corresponding to a level of no motion, $\delta T$ and $\delta S$ are temperature and salinity increments respectively, $\alpha$ and $\beta$ the expansion and contraction coefficients. The correlation between Sea Level Anomaly (SLA) from altimetry and the dynamic height anomaly computed from in-situ measurements is high for regions deeper than 1000 m (Dhomps *et al.*, 2011), at synoptic, seasonal and interannual time scales.

OceanVar2 allows the application of a more complex linear $\mathbf{V}_\eta$ which derives from the steady state results of a linear barotropic model forced by buoyancy anomalies induced by the temperature and salinity increments (DB08). The barotropic model equations, discretized in time by the semi-implicit scheme (Kwizak and Robert, 1971), are:

$$\frac{U^{n+1}-U^{n-1}}{2\Delta t} - fV^n = -gH\frac{\partial\eta^*}{\partial x} - \int_{-H}^{0}\left[\int_{-z}^{0}\frac{\partial(\delta b)}{\partial x}dz'\right]dz + \gamma\nabla^2 U^* \tag{11}$$

$$\frac{V^{n+1}-V^{n-1}}{2\Delta t} + fU^n = -gH\frac{\partial\eta^*}{\partial y} - \int_{-H}^{0}\left[\int_{-z}^{0}\frac{\partial(\delta b)}{\partial y}dz'\right]dz + \gamma\nabla^2 V^* \tag{12}$$

$$\frac{\eta^{n+1}-\eta^{n-1}}{2\Delta t} + \left(\frac{\partial U^*}{\partial x} + \frac{\partial V^*}{\partial y}\right) = 0 \tag{13}$$

where $U$ and $V$ are vertically integrated velocity components, $f$ is the Coriolis parameter, $g$ acceleration due to gravity, $H$ the bottom depth, $\eta$ the surface elevation, $\delta b$ the buoyancy anomaly, and $\gamma$ is the horizontal viscosity coefficient. The superscripts indicate the time step relative to $n$, and the superscript "*" indicates the weighted average between two timesteps. A more detailed description of the barotropic model and its discretization can be found in DB08. In the present version the barotropic



model assumes closed lateral boundaries, thus the possibility to be used as balance operator needs to be carefully evaluated.
In the future the barotropic model will be developed also with open boundary conditions.

In the previous equations, the buoyancy forcing term is defined as:

$$\delta b = g \left( \frac{\delta \rho}{\rho_0} \right)$$

and the density perturbation $\delta \rho$ is estimated by the linear equation:

$$\delta \rho = \alpha \delta T - \beta \delta S \tag{14}$$

Expansion and contraction coefficients ($\alpha$ and $\beta$) in eqs. 10 and 14 can be assumed space independent or spatially variable and estimated linearizing the equation of state around a user defined background temperature ($T_b$) and salinity field ($S_b$):

$$\alpha = \frac{\partial \rho}{\partial T} \big|_{T=T_b, S=S_b}$$

$$\beta = \frac{\partial \rho}{\partial S} \big|_{T=T_b, S=S_b}$$

In the latter case the coefficients are read from an external input file.

In OceanVar2 the sea-level operators produce the final sea surface height increments, replacing the increment produced by the cross-covariance between temperature, salinity and sea-level provided by the EOFs (DB08, Storto *et al.*, 2018).

The choice on the sea level operator has consequences on the velocity operator $\mathbf{V}_{u,v}$. $\mathbf{V}_{u,v}$ computes the velocity correction assuming geostrophic balance under Boussinesq and incompressible approximations:

$$f u_g(z) = -\frac{1}{\rho_0} \frac{\partial p}{\partial y}$$

$$f v_g(z) = +\frac{1}{\rho_0} \frac{\partial p}{\partial x}$$

Decomposing the pressure $p$ at any level $z$ as:

$$p(z) = p_{atm} + g \rho_0 \left( \eta + \int_{-z}^{0} \frac{\delta \rho}{\rho_0} dz \right)$$

where $p_{atm}$ is the atmospheric pressure, $g$ is the effective gravity, $\eta$ is the free surface elevation and $\delta \rho$ is the density departure
from a reference state $\rho_0$. Neglecting the atmospheric pressure, and rewriting the hydrostatic term as buoyancy term, the geostrophic velocities become:

$$f u_g(z) = - \left( g \frac{\partial \eta}{\partial y} + \int_{-z}^{0} \frac{\partial \delta b}{\partial y} dz \right) \tag{15}$$

$$f v_g(z) = + \left( g \frac{\partial \eta}{\partial x} + \int_{-z}^{0} \frac{\partial \delta b}{\partial x} dz \right) \tag{16}$$

When adopting the barotropic model as $\mathbf{V}_\eta$, the sea surface height in eqs. 15 and 16 is replaced by the increments deriving
from the solution of the barotropic model and the lower limit of baroclinic term integral reaches -*H(x,y)*, the ocean floor. On the other hand, when $\mathbf{V}_\eta = \mathbf{V}_{DH}$ the horizontal pressure-gradient force must vanish at the level of no motion, D, in (10). In eqs. 15 and 16 $\eta$ is then substituted with $\delta \eta^{DH}$ and the velocity increments are computed only up to the depth D. Thus, in case of the barotropic model, velocity corrections are provided for the entire water column, while in case of dynamic height velocity corrections are provided from surface to the level of no motion.





## 3.2. The observational error covariance matrix and quality checks


The observation error $\epsilon^0$ is defined as the difference between the observation vector $y$ and the observation counterpart in the true state $x^t$ (Brankart $et$ $al.$, 2009)

$$y = h^*[x^t] + \epsilon^0 \quad \rightarrow \quad \epsilon^0 = (y - y^t) - (h^*[x^t] - y^t) = \epsilon^m + \epsilon^r$$

where $y^t$ is the $true$ (unknown) observation value, $h^*$ is an observation operator, $\epsilon^m$ labels the instrumental or measurement
error (distance of the actual value from the true state) and $\epsilon^r$ gathers the different components of the representativeness errors, due to inaccuracies in $h^*$ and the sampling error of the observations with respect to the true signal. Under the assumption of unbiased error $< \epsilon^0 > = 0$ and that $\epsilon^m$ and $\epsilon^r$ are uncorrelated, the error covariance matrix $\mathbf{R}$ can be constructed as the sum of two terms that can be estimated independently :

$$\mathbf{R} = < \epsilon^0 \epsilon^0 > \approx < \epsilon^m \epsilon^m > + < \epsilon^r \epsilon^r > = \mathbf{R}^m + \mathbf{R}^r .$$

If the errors associated to different observations are uncorrelated, the two matrices greatly simplify in diagonal ones. This hypothesis is valid for most of current global/regional observational data set and it is generally correct when observations are sampled relatively far in time (say few hours to avoid cross-correlation term in $\mathbf{R}^m$) or sparse with respect to grid resolution (to not include off-diagonal elements in $\mathbf{R}^r$). Observation errors are a function of observation type in OceanVar2. Several options are implemented in the OceanVar2 to shape the observation error and the interested reader is asked to consult with the
code manual, available with the code.

OceanVar2 also contains various procedures for the quality control of observations. A background quality check is included to reject observations that are too far from the model estimate. This quality check uses a threshold on the squared misfit defined in (2). Alternatively, OceanVar2 allows a relaxation of the Gaussian approximation when large initial misfits are involved. In standard theory, these misfits are associated with large weights and minimisation is primarily in the direction of reducing such
innovations rather than reducing the innovations close to zero. In OceanVar2, following Storto (2016), the initial misfit distribution can be approximated with a Huber norm PDF to reduce the impact of such tails.

Provided that observation errors are assumed to be spatially and temporally uncorrelated, horizontal and vertical data thinning rejects observations too close in space. In case of multiple data from the same instrument falling in the same model grid cell, only the observation closest to the analysis time is retained. Coastal rejection can prevent the assimilation of altimetric and in-
situ coastal observations, to avoid inconsistencies between observed and modelled coastal processes. In addition, a rejection criterion based on the model bathymetry can be activated preventing the assimilation of data in shallow areas.

## 4. Experimental Design

The experimental design is driven by the aim of showing the best set up of OceanVar2 for the assimilation of satellite altimetry together with ARGO floats and XBT in the Mediterranean Sea. The set-up of the ocean model used in this study is a simplified
version of the physical component of the Mediterranean Forecasting system of the Copernicus Marine Service (Clementi $et$





*al.,* 2023). The model is implemented over the entire Mediterranean basin (Fig.1) with a horizontal grid resolution of 1/24˚ (approx. 4 km) and 141 non-uniformly distributed vertical levels. The ocean model code is based on the Nucleus for European Modelling of the Ocean NEMO (v4.2, Madec *et al.*, 2023) and includes the representation of tides. Atmospheric forcings are calculated interactively with the operational fields of the European Centre for Medium-Range Weather Forecasts (ECMWF).

The only difference to the Copernicus operational system is the omission of the surface wave modelling coupling. Details on the model implementation can be found in Clementi *et al.* (2023).

Starting from an operational analysis, we performed a 1-year simulation followed by one year of daily assimilation cycles of in-situ (XBT and ARGO floats) and satellite SLA data for the whole 2021 year. Fig.1 shows the positions of the assimilated in-situ and SLA data; the SLA data refer to a period of 21 days. In our experimental set-up we perform daily assimilation

cycles starting at midnight every day and we assimilate all the data available the day previous the analysis time.

### 4.1. Correcting the misfits for tides

A fundamental aspect to consider when assimilating SLA is the possible presence of tides in the modelled solution and in the observed data. Discrepancies between modelled and observed tides can, as a first approximation, be attributed to inaccuracies in the bathymetry of the model, the bottom and/or the coastal frictional dynamics. However, the misfit containing tidal signal

between observed and modelled estimates in the present OceanVar2 formulation would be projected into baroclinic increments by the covariance matrix of the background error (eq. 8). It is therefore essential to filter out the tidal signal from both the observed and modelled SLA. This paper offers a solution to the assimilation of satellite altimetry in a model with tides, showing that a filtering procedure can be accurate enough and that no additional adjustment is required in the analysis.

The Copernicus along-track sea level anomalies are provided together with an estimate of the tidal signal along the tracks so

tides can be filtered easily from the observations. To remove the tidal signal from the model background field, the tidal amplitude and phase for the eight components included in the Mediterranean Sea model (M2, S2, K1, O1, N2, P1, Q1 and K2) have been derived from a simulation output by harmonic analysis of the hourly sea level field. Following Cao *et al.* (2015), six months of hourly data were used for the harmonic analysis. The harmonic analysis was performed using Pawlowicz *et al.* (2002) algorithm, based on the Foreman method (Foreman 1977; Foreman 1978) at each model grid point. Knowing the tidal

constants, it is possible to estimate the model tidal sea level at the exact time and location of the altimetry data and remove this component from the model outputs.

During the model simulation, misfits between model estimates and observations are computed and before entering the OceanVar2, the misfits are updated removing the tidal signal from both observations and model results. In Fig.2 an example of SLA satellite track is provided with model estimates and the satellite observations, the position of the track is shown in

Fig.1. In Fig.2 upper panel, the full signal from the model simulation and the observations is drawn as a function of latitude along the track. Figure 2 middle panel shows the de-tided signals in addition to a de-biasing procedure described by Dobricic *et al.* (2012). Dobricic *et al.* (2012) shows that this method is the best for considering differences between the large-scale steric signal and the mean dynamic topography between observations and model. The average difference along the track is removed





if the track is continuous, or for individual segments if the track is discontinuous due to the presence of land. Finally in Fig.2
bottom panel the two tidal components for the observational and modelled SLA are shown indicating the large-scale signal of
tides in the open ocean.

## 4.2. Sensitivity experiments

In addition to a simulation, two sets of 1-year assimilative experiments are presented and their results compared and assessed
against each other and observations. All the experiments assimilate ARGO floats and XBT temperature and salinity data in the
whole domain including the Atlantic part, while SLA data are assimilated only within the Mediterranean Basin (see Fig.1). In
every experiment the vertical component of the background error covariance matrix is modelled using 25 tri-variate EOFs
(temperature, salinity and SLA) computed following Dobricic *et al.* (2006) for every model grid point. The EOFs are computed
from a 30 years timeseries of the Mediterranean Sea reanalysis (Escudier *et al*., 2021). The horizontal correlation radius was
set to a constant value of 27 kilometres, determined through sensitivity experiments. The diffusive filter was iterated five times
to model the horizontal covariance. To account for coastal effects, the correlation radius was linearly decreased starting from
30 kilometres offshore to the minimum grid resolution near the coast. Additionally, a Neumann boundary condition was
applied at the coast, setting the normal derivative of the field to zero. Observations are rejected if they are closer than 15 km
from the coast and if the misfits are larger than fixed thresholds: 5ºC for temperature, 2 psu for salinity; and 30 cm for SLA.
The observational error covariance matrix is assumed diagonal. All the SLA data have an associated error of 3 cm regardless
of the satellite and the geographic distribution. The observational errors for in-situ observations were tuned via the Desroziers'
method (Desroziers *et al*., 2005) and varies monthly. Temperature and salinity observational errors peak at the surface with
values of 0.45 ºC and 0.14 psu, from 75 to 325m depth they decrease linearly to values of 0.2 ºC and 0.05 psu, starting from
750m they have constant values of 0.1 ºC and 0.02 psu respectively.

We performed two sets of experiments with different OceanVar2 sea level operators and choices of free parameters. In Exp-
1, which is used as a reference experiment, the barotropic model is used as sea level operator with constant (in space and time)
α and β in eq.14 and SLA data are rejected when falling in areas shallower than 100m. In the second experiment (Exp-2),
consistently with Adani *et al.* (2011), we rejected SLA data falling in areas shallower than 150m. Experiment 3 is similar to
Exp-2, but we test the sensitivity to variable expansion and contraction coefficients in eq.14. The coefficients are computed
linearizing the equation of state around a monthly mean climatology. In all these three experiments we integrated the barotropic
model for 3 days with a time-step of 3600 sec and then used the average of the last day as approximation of the steady state
solution, the integration of the barotropic model is fully implicit and the turbulent viscosity is equal to 650 m2/s.

Experiments 4, 5 and 6 use the dynamic height as sea level operator. The difference among them is the choice of the level of
no motion depth. In Exp-4 we used a level of no motion equal to 150m, thus Exp-2 and Exp-4 differ only for the sea level
operator. In Exp-5 the level of no-motion is 350m in agreement with the Mean Dynamic Topography (Rio *et al.*, 2014) used.
Finally, in Exp-6 the level of no motion is set at 1000 m which is the traditional choice for the operational setting of the
Mediterranean Sea forecasting system, Coppini *et al.* (2023). In all these experiments, the depth of the level of no motion



naturally coincides with the minimum depth of SLA observations inclusion in the data assimilation scheme. In the first set of experiments, even if the state vector contains also the velocity field, the increments are applied only to the assimilated variables ($T$, $S$ and $\eta$). In the second set of experiments, also velocity corrections are applied. Throughout the remainder of the paper, 320   experiments marked with an asterisk refer to those with velocity corrections. The first sets of experiments are summarized in Table 1: the second set of experiments has the same naming convention as the first set, with the sole difference being the application of the velocity correction in the analysis definition.

|  | **Exp-1** | **Exp-2** | **Exp-3** | **Exp-4** | **Exp-5** | **Exp-6** |
|---|---|---|---|---|---|---|
| **Sea level operator** | BM | BM | BM | DH | DH | DH |
| $\boldsymbol{\alpha/\beta}$ | Const. | Const. | MC | Const. | Const. | Const. |
| **Reference Level for DH (m)** | N/A | N/A | N/A | 150 | 350 | 1000 |
| **SLA Min depth rejection (m)** | 100 | 150 | 150 | 150 | 350 | 1000 |

**Table 1** Sensitivity experiments set-up. First row indicates the sea level operator used: BM= Barotropic Model; DH=Dynamic height. Second 325   row indicates the choice for expansion and contraction coefficients: constant in space and time (Const) and spatially and temporally variable as computed from monthly climatology (MC). Third row indicates the reference level for the lower integral limit of the dynamic height operator and thus the level of not motion for the $\mathbf{V_{u,v}}$ part of the model background error covariance matrix. Fourth row indicates the minimum depth used as criterium to reject SLA data.

## 5. Results

Before analysing the skills of the different experiments, temperature and salinity increments obtained, starting from the same set of SLA misfits, but using the different assimilation scheme set-up described in Table 1 are shown in Fig.3. The differences between the experiments are generally small, and of the order of 10%. The largest differences are due to the different number of SLA data assimilated, because of the different level of no motion used or the different minimum depth rejection criterion adopted. We note that when the same data are assimilated, thus in areas deeper than 1000m, very similar increments in SLA 335   are generated by the OceanVar2 regardless of the schemes adopted. However, the schemes differ on how these increments are projected into temperature and salinity increments. Note the ordinate axes are strongly stretched in the figure to highlight the first 150m depth where most of the corrections are confined. Comparing Exp-2 and 3, which differ only in the use of spatially and temporally variable expansion and contraction coefficients, we observed small but noticeable differences in the temperature and salinity increments, particularly in the amplitude of near-surface maxima. The choice of sea level operator 340   substantially influenced the results. When using Dynamic Height with a level of no motion set at 1000 meters (Exp-6), temperature and salinity increments were comparable to those obtained with the barotropic model. The primary cause of the observed differences appears to be the constraint imposed on assimilated data by the level of no motion. Reducing the level of no motion (Exp-4 and 5) allowed for the assimilation of more sea level anomaly data, but resulted in considerably different temperature and salinity increments within the first 100 meters compared to the barotropic model.





To fully assess the performances of Table 1 experiments, the mean squared error (e.g., Murphy, 1988) is decomposed and the single components analysed:

$$MB = \bar{m} - \bar{o}, \tag{17}$$

$$SDE = \sigma_m - \sigma_o, \tag{18}$$

$$CC = \frac{1}{\sigma_o} \frac{1}{\sigma_m} \frac{1}{N} \sum_{i=1}^{N} (m_i - \bar{m})(o_i - \bar{o}), \tag{19}$$

where MB is the mean bias error, SDE is the standard deviation error and CC is the cross correlation between the modelled and observed fields. The *i-th* modelled and observed variable is denoted by $m_i$ and $o_i$, respectively; $\bar{m}$ and $\bar{o}$ are the respective averages (horizontal and temporal); while $\sigma_m$ and $\sigma_o$ are the respective standard deviations. In addition, the unbiased root mean squared error (*uRMSE*) is computed:

$$uRMSE = \sqrt{\frac{1}{N} \sum_{i=1}^{N} [(m_i - \bar{m}) - (o_i - \bar{o})]^2} \tag{20}$$

It is important to note that the model results and observations used here are the same as those used to calculate misfits within the assimilation cycle. However, while not all misfits are utilized in the assimilation process, all available observations are included in the error statistics. This ensures that all experiments are evaluated based on the same reference dataset of observations. Furthermore, to evaluate model performance even in very shallow regions, the observational dataset used in the misfits, and thus in the calculation of the error statistics, includes sea level anomaly (SLA) data covering regions up to 10m

depth. This allows for the assessment of model skill in very shallow regions where data are not assimilated in any of the presented experiments.

### 5.1. Barotropic sea level operator and simulation comparison

The performance of Exp-1 is compared with the non-assimilative model simulation. In Fig.4 the statistics for the SLA are

shown. Every point in the time-series represents 5-day window statistics. That is, the overbars and the standard deviations in eqs.17, 18, 19 and 20 are computed over a 5-day time window.

The simulation has an error, slightly growing during the second half of the year, of about 5 cm. The model with assimilation underwent a 10-20 day adaptation period, after which the *uRMSE* of the misfit stabilizes around 3 cm. A slight but consistent improvement is noted in the *CC*. No seasonal cycle is observed in the CC of Exp-1, whereas the simulation exhibits a distinct

summer minimum in the correlations. The *SDE* in the simulation is generally negative and it is characterized by 5-days oscillations. In Exp-1, the SDE (Fig.4 bottom panel) stabilized around values of 0.25 cm, indicating an overestimation of the observed variability.

The SLA yearly averaged statistics were clustered according to ocean depth and are plotted in Fig.5. In areas with bathymetry between 150 and 2500 m, the simulation exhibited an almost constant *uRMSE*. However, the error increases in shallower and

deeper regions, reaching the maximum in areas deeper than 3500 m. The *uRMSE* of the Exp-1 was more constant and





approximately half of the corresponding simulation statistics. Regardless of the region considered, Exp-1 has better statistics than the simulation, indicating the effectiveness of the assimilation procedure. For the *CC* differences between simulation and Exp-1 results are also evident. In the simulation, *CC* decreases with depth while for the Exp-1 we observe an opposite tendency. In terms of *SDE*, the largest improvements w.r.t. the simulation are confined in shallow areas. The simulation underestimates

the variance in areas with bathymetry shallower than 1000m, this is particularly evident in areas shallower than 150m. The model with data assimilation tend to overestimate the observed variability particularly in shallow areas.

In Fig.6 the vertical profiles of *uRMSE*, *MB*, *CC* and *SDE* for salinity (upper panels) and temperature (bottom panels) for the simulation and the Exp-1 are shown. Statistics are computed against all available ARGO and XBT profiles.

In Fig.6 the vertical structure of the salinity *uRMSE* is similar between the simulation and the Exp-1. The salinity errors are

characterized by a near surface maxima which is reduced in the assimilative run. The *MB* in the simulation has a subsurface minimum at 100 m depth, while Exp-1 misfits have almost homogenous values through all the water columns. The *CC* resembles the vertical distribution of the *uRMSE*, with values approaching the unity in both the experiments below 500 m depth. Finaly the salinity *SDE* confirms the large improvement arising from the assimilation procedure. In the near surface layer, the simulation and the assimilative run have opposite behaviours, with the simulation overestimating the observed

variability while the assimilative run underestimating it. Below 100 m depth the *SDE* salinity values are noticeably reduced.

The temperature *uRMSE* and *CC* are characterized by a strong, summer intensified (not shown), subsurface maximum/minimum due to the model difficulties in reproducing the correct stratification. A second *uRMSE* maximum (*CC* minimum) is present around 300 m probably related to the misrepresentation of the Levantine Intermediate Water (LIW) advection in the different Mediterranean regions. A third temperature error relative maximum is present between 1000 and

1500 m depth. The assimilation corrects all the errors by approximately 30-50% down to 500 m, less below this depth.

Temperature *MB* is largely improved by assimilation. The simulation tends to overestimate the observed temperature variability, and the *SDE* has a marked vertical structure. In general, the assimilation seems capable to correct most of the model errors except in the upper thermocline/mixed layer depth. Analysis of the corresponding time-series (not shown) indicates a clear summer maximum in all the error statistics in proximity of the mixed layer depth. This behaviour is shared between all

the different experiments, but it is clearly reduced in the assimilative runs.

**5.2 Sensitivity experiments to the sea level operator**

Given that the *CC* is always positive for all the experiments, the misfit statistics for the experiments listed in Table 1 are analysed in terms of relative improvement with respect to the simulation or Exp-1 according to the following metrics definition:

$$S\_uRMSE_{Exp\#} = \left( \frac{uRMSE_{Ref} - uRMSE_{Exp\#}}{uRMSE_{Ref}} \right) \times 100$$

$$S\_CC_{Exp\#} = \left( \frac{CC_{Exp\#} - CC_{Ref}}{CC_{Ref}} \right) \times 100$$



Where # indicates the different experiments listed in Table 1 and "*Ref*" the simulation or Exp-1. In the following these statistics are presented only for the SLA data given that a similar comparison for the temperature and salinity did not provide additional insights.

Figure 7 illustrates the performance of the six different data assimilation experiments in terms of improvements in root-mean-

square error (*S_uRMSE*) and correlation coefficient (*S_CC*) relative to the model simulation. The top panels display the time evolution of these improvements, highlighting both short-term fluctuations and overall trends. The bottom panels present the time-averaged *S_uRMSE* and *S_CC* improvements clustered by bathymetric depth ranges, revealing how the effectiveness of each experiment varies with depth. All assimilative experiments outperformed the model simulation. Experiments using the dynamic height as the sea level operator, with levels of no motion set at 150 meters (Exp-4) or 1000 meters (Exp-6), generally

performed worse than the other experiments. For Exp-6, the differences in both *uRMSE* and *CC* were particularly noticeable in regions shallower than 1000 meters, where SLA data were not assimilated. However, even in these regions, Exp-6 substantially outperformed the model simulation, suggesting that corrections applied in deeper areas effectively propagated into shallower regions. In Exp-4, the deterioration in results compared to other experiments was primarily confined to deeper regions. In areas shallower than 150 m Exp-1 outperforms the other experiments, however the performances of the experiments

are similar indicating that the coastal areas are strongly constrained by the open ocean dynamics. A clear dependence of the *S_CC* on model bathymetry was evident in all experiments. The percentage of improvement in *S_CC* increased with growing depth, with the most significant improvements observed in areas deeper than 3500 meters, where the model simulation exhibited the smallest *CC*. In shallow regions, Exp-6 generally provided the smallest improvement in *S_CC* compared to the other experiments. The results demonstrate that certain experiments achieve substantial improvements in deep-ocean regions,

while others show more consistent performance across all depths. These results highlight the challenges associated with choosing an appropriate level of no motion in data assimilation of SLA. The choice of the level of no motion can significantly impact the accuracy of model analysis, especially in complex regions with varying bathymetry and ocean dynamics.

Figure 8 presents the relative performance of five data assimilation experiments (Exp-2 through Exp-6) compared to a baseline assimilative experiment (Exp-1), now used as the reference. The top panels illustrate the time series of percentage changes in

root-mean-square error (*S_uRMSE*) and correlation coefficient (*S_CC*). The bottom panels depict the time-averaged *S_uRMSE* and *S_CC* changes, categorized by bathymetric depth ranges. In contrast to the previous figure, where improvements were relative to a model simulation, this figure demonstrates the relative performance of each experiment against the initial data assimilation run. Negative values indicate a decrease in performance (higher *uRMSE* or lower *CC*) compared to Exp-1, while positive values indicate improvement. This comparison highlights the incremental benefits or drawbacks of different

experimental setups in relation to a specific data assimilation configuration.

All the experiments employing the barotropic model perform similarly to each other. In terms of time-series comparison, Exp-4 (with dynamic height as sea level operator and level of no motion equal to 150m) has performance worse than all the other experiments. Among the experiments using the dynamic height operator, Exp-5 generally has better results both in terms of *uRMSE* and *CC*. The analysis per bathymetric classes shows  better the differences among the experiments. None of the





experiments outperform Exp-1 in regions shallower than 150m both in terms of *uRMSE* and *CC*. In deeper areas we see that the Exp-4 and Exp-5 produce, in general, worse results, and the worsening is amplified as the depth increases and the level of no motion decreases; 1000m depth is a clear boundary for the effectiveness of Exp-6. Employing the barotropic model as a sea level operator yields consistent results, with small sensitivity to the minimum depth used in the rejection criterion or to the choice of constant or variable expansion/contraction coefficients. This confirms the difficulty of establishing a constant level

of no motion and highlights the benefit of using the barotropic model as a balancing mechanism. Table 2 provides a summary of the spatially and temporally averaged statistics for all experiments, including those from the second set discussed subsequently.

| | uRMSE | CC | SDE | S_uRMSE$_{sim}$ | S_uRMS$_{exp-1}$ | S_CC$_{sim}$ | S_CC$_{exp-1}$ |
|---|---|---|---|---|---|---|---|
| **Sim** | 5.05 | 0.25 | -0.02 | / | / | / | / |
| **Exp-1** | 2.85 | 0.75 | 0.27 | 43.49 | / | 197.64 | / |
| **Exp-2** | 2.85 | 0.75 | 0.28 | 43.59 | 0.17 | 197.74 | 0.04 |
| **Exp-3** | 2.83 | 0.75 | 0.28 | 43.94 | 0.80 | 199.03 | 0.47 |
| **Exp-4** | 3.02 | 0.72 | 0.24 | 40.18 | -5.85 | 186.08 | -3.88 |
| **Exp-5** | 2.89 | 0.74 | 0.28 | 42.75 | -1.31 | 194.75 | -0.97 |
| **Exp-6** | 2.92 | 0.73 | 0.28 | 42.12 | -2.42 | 192.38 | -1.77 |
| **Exp-1$^*$** | 2.76 | 0.77 | 0.19 | 45.40 | 3.37 | 205.74 | 2.72 |
| **Exp-2$^*$** | 2.77 | 0.76 | 0.21 | 45.15 | 2.94 | 204.52 | 2.31 |
| **Exp-3$^*$** | 2.77 | 0.76 | 0.22 | 45.11 | 2.86 | 204.18 | 2.20 |
| **Exp-5$^*$** | 2.79 | 0.76 | 0.22 | 44.80 | 2.32 | 203.16 | 1.86 |
| **Exp-6$^*$** | 2.84 | 0.75 | 0.22 | 43.63 | 0.24 | 199.13 | 0.50 |

**Table 2.** Global averages of experiments statistics. Units in *uRMSE* and *SDE* are cm. Subscripts in the relative performance statistics [%]
indicate the reference experiment used (simulation or Exp-1).

### 5.3. Sensitivity experiments to velocity corrections

Another set of experiments (Exp-1* to Exp-6*) was carried out including velocity corrections in the analysis estimates. The OceanVar2 setup used in Exp-4* generated velocity increments that led to numerical instabilities in the ocean model,
preventing this simulation from completing. In contrast, the other experiments in this set ran without such issues, underscoring the challenges associated with dynamic height methods. Consequently, our focus is on the stable experiments. Notably, the successful experiments in the second set, which included velocity corrections, demonstrated improved performance compared to their counterparts in the first set, which lacked velocity corrections (Table 2). The extent of improvement varies depending on the specific experiment and the region analysed.

Figure 9 shows time series and Fig.10 presents temporally averaged statistics by bathymetric class for the second set of experiments. In terms of time series (Fig.9), the error components exhibit the same characteristics as those discussed for Exp-





1 (Fig.4 and 5) . The *uRMSE* exhibits a summer minimum in all experiments. Exp-6[*] performs significantly worse than the others throughout the year. All experiments using the barotropic model have similar *uRMSE* values, with Exp-1[*] generally appearing slightly better than the others. The correlation coefficient (*CC*) increases throughout the experiment's length. For this statistic as well, Exp-6[*] is the worst, showing consistently lower values than the other experiments. Even for the SDE, which is generally reduced compared to the previously studied experiments, the relative performance of the different experiments seems to be confirmed.

Figure 10 presents the temporally averaged statistics clustered according to the bathymetry. All the experiments benefited greatly from the inclusion of the velocity corrections. Exp-6[*] confirms its poor performance in areas shallower than 1500m. However, by also correcting the velocities, its statistics in deep areas are now similar, or slightly better, to those obtained from experiments using the barotropic model as operator in the background error covariance matrix. Exp-1[*] is now the best among those analysed for all the bathymetric classes shallower than 500m.

In terms of correlation coefficient, the results previously obtained by analysing *uRMSE* seem to be confirmed. For shallow areas (<1000m), Exp-6[*] is significantly worse than the others. In all other bathymetric classes, even confirming the previous findings, the differences between the experiments are less pronounced. A different behaviour is observed when analysing the standard deviation of the error. Exp-1[*] remains the setup that shows significantly lower error values than the others in almost all bathymetric classes. However, for this statistic as well, the differences between the experiments have decreased compared to the experiments where velocity correction was not applied. Experiment-6[*] is the one that benefits the most from velocity corrections in very deep water.

## 6. Performances and Parallelization

To optimize computational performance, OceanVar2 adopts a domain-decomposition scheme. This scheme leverages the computing power of a parallel computer by partitioning the computational domain into subdomains. Each process executes the necessary operations to update its portion of the global domain, sharing communications with neighbouring processes for lateral boundary treatments using MPI calls (Message Passing Interface).

Rigorous testing has been conducted to guarantee bit-for-bit (BFB) reproducibility across runs with different MPI processes as well as runs with the same amount of MPI processes but different partitioning of the structured geographic grid. The quasi-Newton L-BFGS minimizer (Byrd *et al.*, 1995), employed for numerical minimization of the cost function, necessitates global matrix-vector multiplication, which precludes BFB reproducibility when domain decomposition is utilized. Divergences between executions stem from the non-associativity of floating-point operations, particularly floating-point summation within the minimizer. To mitigate this, OceanVar2 offers the flexibility to execute the minimizer serially while the remaining code is parallelized using MPI domain decomposition. Extensive testing has demonstrated that serial execution of the minimizer, aggregating variables from all domains, ensures BFB reproducibility. Moreover, even when the minimizer is executed in





parallel, differences arising from various domain decompositions are statistically insignificant. Possible future work includes the introduction of a different minimizer suited for MPI parallelization.

Neglecting the differences arising from the parallel execution of the minimizer, the computational performances of the different experiments were evaluated in terms of minimizer iterations and code scalability. Figure 11 compares the number of iterations required for the minimizer to converge in the various OceanVar2 experiments. The results are presented as a probability distribution, with statistics calculated based on the year of assimilation testing.

All experiments using the dynamic height operator converged with fewer iterations than those employing the barotropic model.

The choice of the level of no motion only slightly affected convergence, with a median increase from 24 to 25 iterations when using 350m or 150m instead of 1000m. Schemes with the barotropic model required few more iterations, and the median is 25 for Exp2 and 26 for Exp-1 and Exp-3.

To assess scalability, we limited the comparison to Exp-2 and Exp-4, as they had the same number of assimilated observations. We tested OceanVar2's performance with increasing numbers of cores. For a fixed number of cores, we explored different

minimization strategies (using 8 different set of observations) and various decomposition strategies (e.g., with 16 cores, we tested 4x4 and 8x2). The model grid consisted of 1307x380x141 points along the x, y, and z directions, respectively. Results are shown in Fig.12. Up to 36 cores, the experiment with the dynamic height operator consistently outperformed the one using the barotropic model. However, Exp-4 reached a performance plateau at 36 cores, while Exp-2 demonstrated a slight improvement up to 72 cores, where the performance of the two setups became identical. For a larger number of processors,

we observed a deterioration in performance due to increased communication load.

## 7. Conclusions

This study describes recent developments of the OceanVar variational ocean data assimilation scheme. Key innovations compared to the previous schemes (DB08, Storto *et al.*, 2011, 2014) include the implementation and evaluation of two alternative solutions for the sea level operator, encompassing both barotropic model and dynamic height operator. Furthermore,

a diffusive operator has been adopted to model Gaussian horizontal covariances, replacing the recursive filter used in previous code versions. Finally, the geostrophic velocity operator is utilized for total velocity corrections, deviating from the DB08 approach and applied to both dynamic height and barotropic sea level operators.

Furthermore, a method for filtering the tidal components of the background model fields is applied and tested allowing the assimilation of de-tided SLA, together with in-situ temperature and salinity data to produce analyses. These OceaVar2 new

and old features have been tested and compared for a regional implementation of the assimilation scheme in the Mediterranean Sea.

It has emerged that the barotropic operator is the only one capable to consistently assimilate sea level anomaly data in shallow and deep ocean regions. Variable *alpha* and *beta* parameters in the linear equation of state yielded minor differences in our experiments, however this assumption is likely not to be valid in global models.



The dynamic height operator, though easy to implement, has clear limitations. Requiring the definition of a spatially independent level of no motion, it does not provide an optimal solution in domains with highly variable bottom topography and dynamics. For the Mediterranean Sea, a level of no motion equal to 1000m is appropriate, as demonstrated by the quality of the corrections obtained with OceanVar2. However, this represents a significant limitation, as it excludes the assimilation of SLA observations in shallower areas. Decreasing the level of no motion depth reveals the limitations of this approach. For

shallower levels, the benefits of assimilating more data are offset by the loss of the quality of the corrections in deeper areas. The results are corroborated by the numerical instabilities arising when velocity corrections are applied in experiments with a level of no motion shallower than 350m.

Computationally the barotropic model is more expensive than the dynamic height operator, however it has a minor impact on the minimization iterations. Further, the semi-implicit scheme used to discretize the barotropic equations allows for large time-

step significantly limiting the computational demand. The adopted solutions simplify the application of the OceanVar2 in complex areas of the world ocean. To our knowledge OceanVar2 is the only data assimilation scheme employing a barotropic model in its model background error covariance matrix. It's important to note that the current implementation of the barotropic model uses closed lateral boundary conditions. Its applicability is therefore limited to basins with a geometry that allows this approximation. The OceanVar2 code is stable, robust, its previous versions have been largely documented in several scientific

papers, and the present version is also open to the community. Future developments could explore the implementation of lateral open boundary conditions into the barotropic model, interfacing the system with AI, and adapting it to unstructured grids and/or global applications.

**Code availability**

The OceanVar2.0 code is publicly available under a GPLv3 licence (https://www.gnu.org/licenses/gpl-3.0.txt) at https://github.com/CMCC-Foundation/OceanVar2 (this manuscript) together with a user guide on compiling and running the code (Adani *et al.*, 2025, https://github.com/CMCC-Foundation/OceanVar2/blob/main/doc/OceanVar_User_Manual.pdf). The code used in this paper is permanently archived at https://doi.org/10.5281/zenodo.15593468 (Oddo *et al.*, 2025). A test case can be downloaded at https://github.com/CMCC-Foundation/MedFS831. The ocean model used is based on the NEMO

source code (version 4.2.0) is accessible Zenodo. https://doi.org/10.5281/zenodo.6334656 (Madec et *al.*, 2022).

**Author contribution**

OP is the main author; he is the lead developer of the OceanVar2. MA played a central role in the discussion; he led the

developments; he wrote the code, and he performed all the experiments. FC contributed to the MPI parallelization and debugging of the OceanVar2 code. AC contributed to the writing of the manuscript, and in incorporating some routines from previous code version. ACG computed the tidal constants used in all the experiments.  EJ, AA, FM and IE were involved in the discussions and the definition of the development strategy. JP computed the EOFs used in all the experiments. EC and SM



played a central role in the discussion. NP contributed to the writing of the manuscript; she was active co-leading the scientific

development.



# Figures

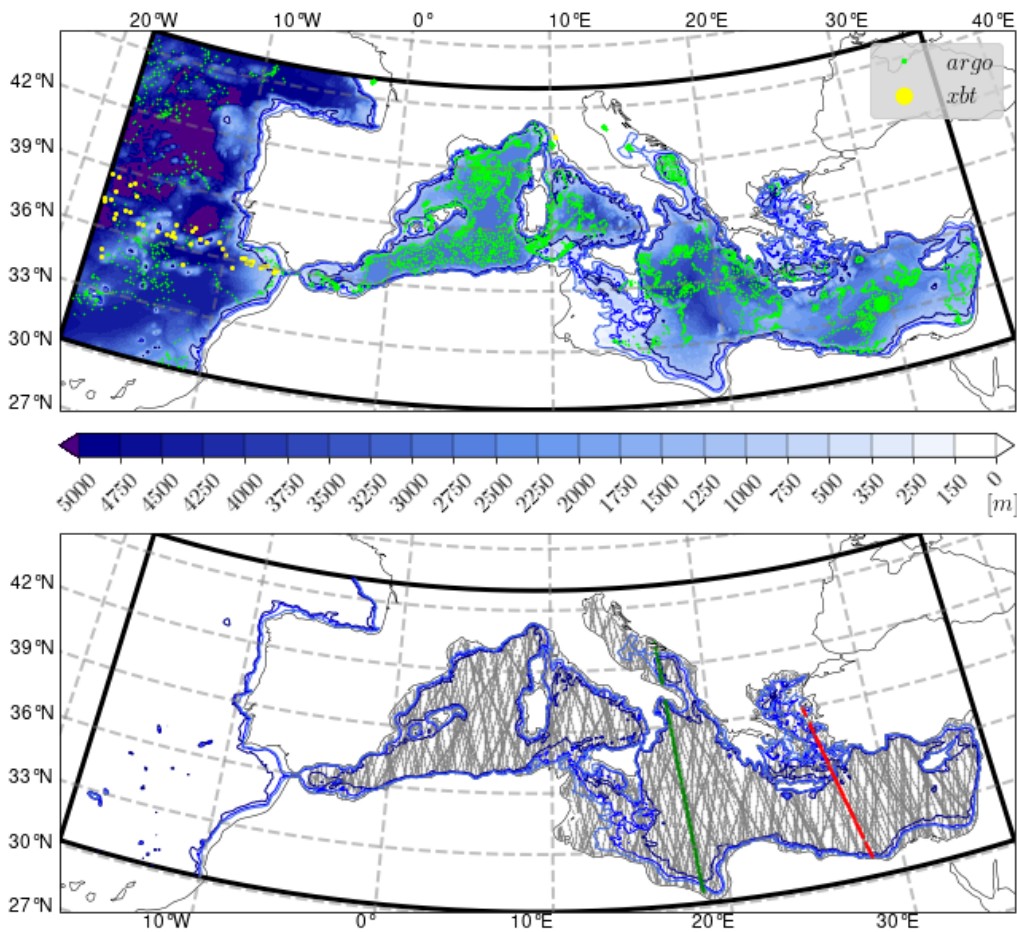

**Fig.1** Top panel: Model domain and bathymetry. Green and yellow dots indicate the position of the assimilated ARGO floats and XBT respectively. Bottom panel: example of 21-day altimetry data. Satellite tracks in red and green are used in Figure 2 and 6 respectively. Three isobaths are drawn in both panels: 150, 350 and 1000m.






**Fig.2** Sea Level Anomaly data example along the red track of Figure 1. Blue lines indicate model simulation results, while black lines indicate observational data. In the upper panel the full signals are plotted. The dashed line indicates where SLA are in regions shallower than 1000m. Middle panel is after the removal of the tidal signals, separately in the model and observations, and the along track averaged difference. Bottom panel shows the along track observational and model tidal signals.





**Fig.3** SLA, Temperature and Salinity increments obtained from the different OceanVar2 set-up listed in Table1, starting from
the same misfits. The SLA track used is drawn in green in Fig.1. For each experiment in the top panel there are the SLA
increments where black dots indicate assimilated data; green dots indicate data rejected based on the coastal distance criteria,
red dots indicate data rejected due to the level of no motion or minimum depth (in case of the barotropic model).

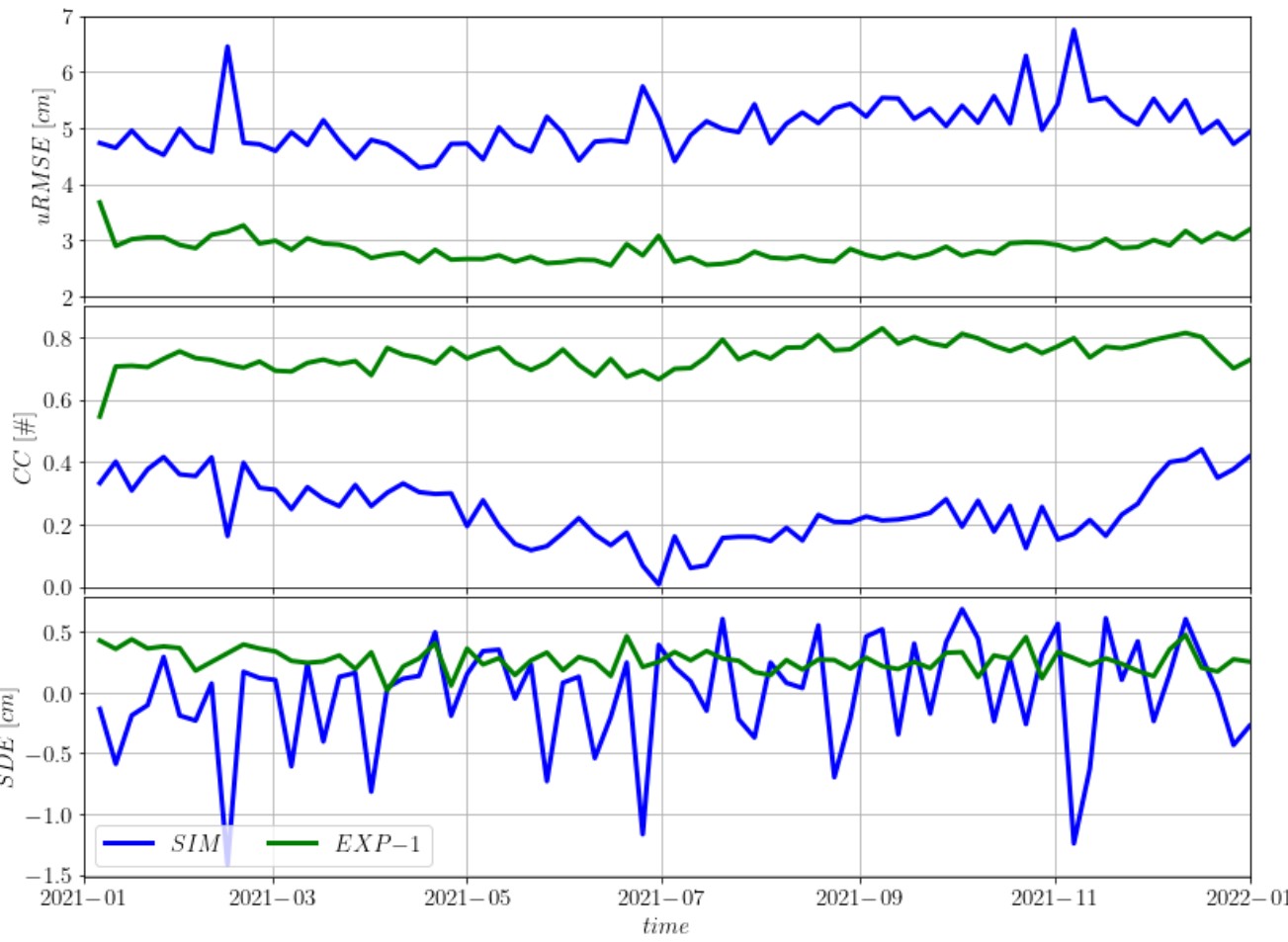

**Fig.4** Time series of SLA error statistics for the entire 2021 year. Blue and green lines indicate the statistics for the simulation results and the misfit of Exp01 respectively. Top panel: unbiased root-mean-square error. Middle panel: correlation coefficient, in the bottom panel the standard deviation error is plotted.







**Fig.5** SLA statistics as a function of the ocean depth. Blue and green bars indicate the simulation and Exp-1 results respectively. In the top panel the number of observations used is also provided with dark bars.





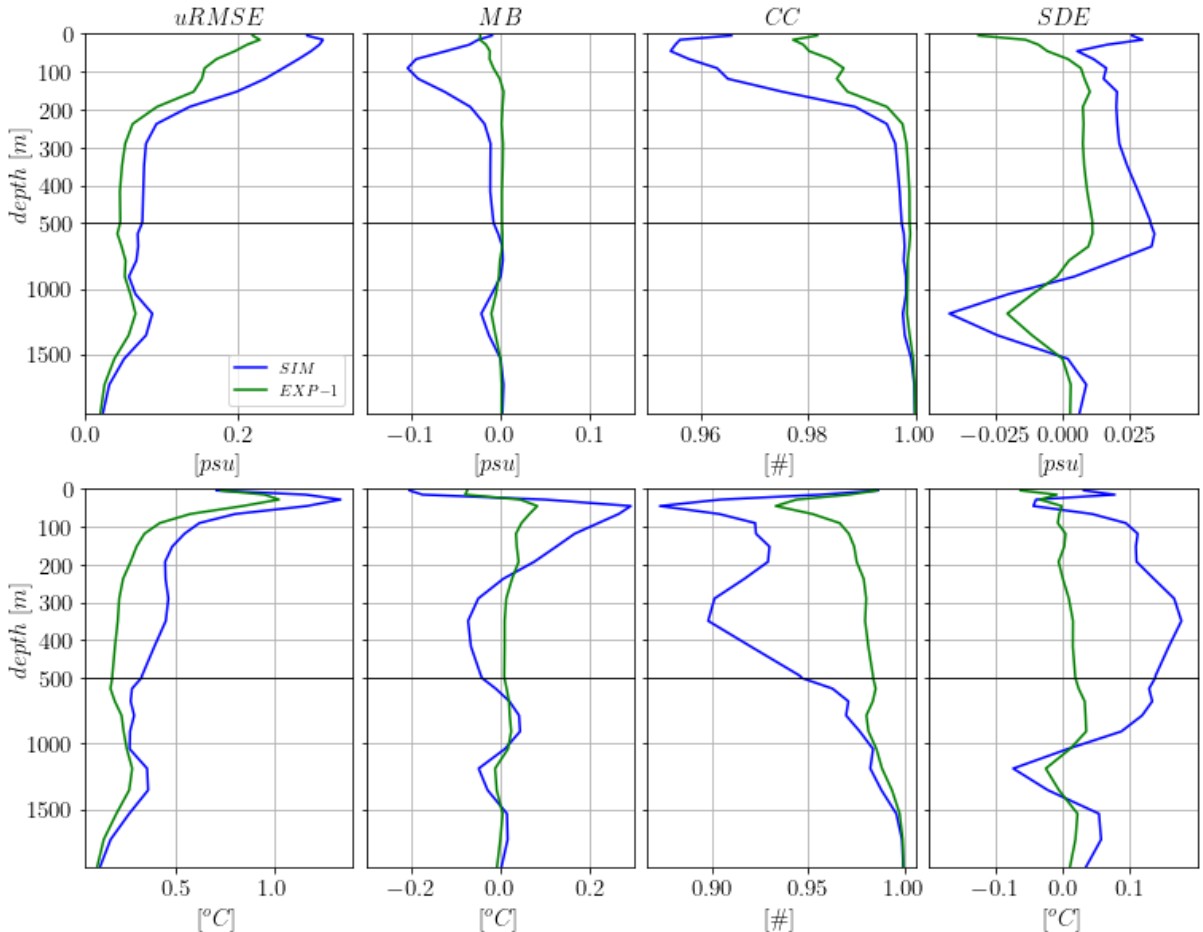

**Fig.6** Vertical profiles of yearly averaged misfit statistics for salinity (top panels) and temperature (bottom panels). From left

to right: *uRMSE*; *ME*; *CC*; *SDE*. Blue and green lines indicate simulation and Exp-1 results respectively.



**Fig.7** *S_uRMSE* and *S_CC* indices for percentage of improvement of the SLA misfit errors for the different experiments w.r.t the simulation. Top two panels: time-series. Bottom panels: average indices for bathymetry classes.


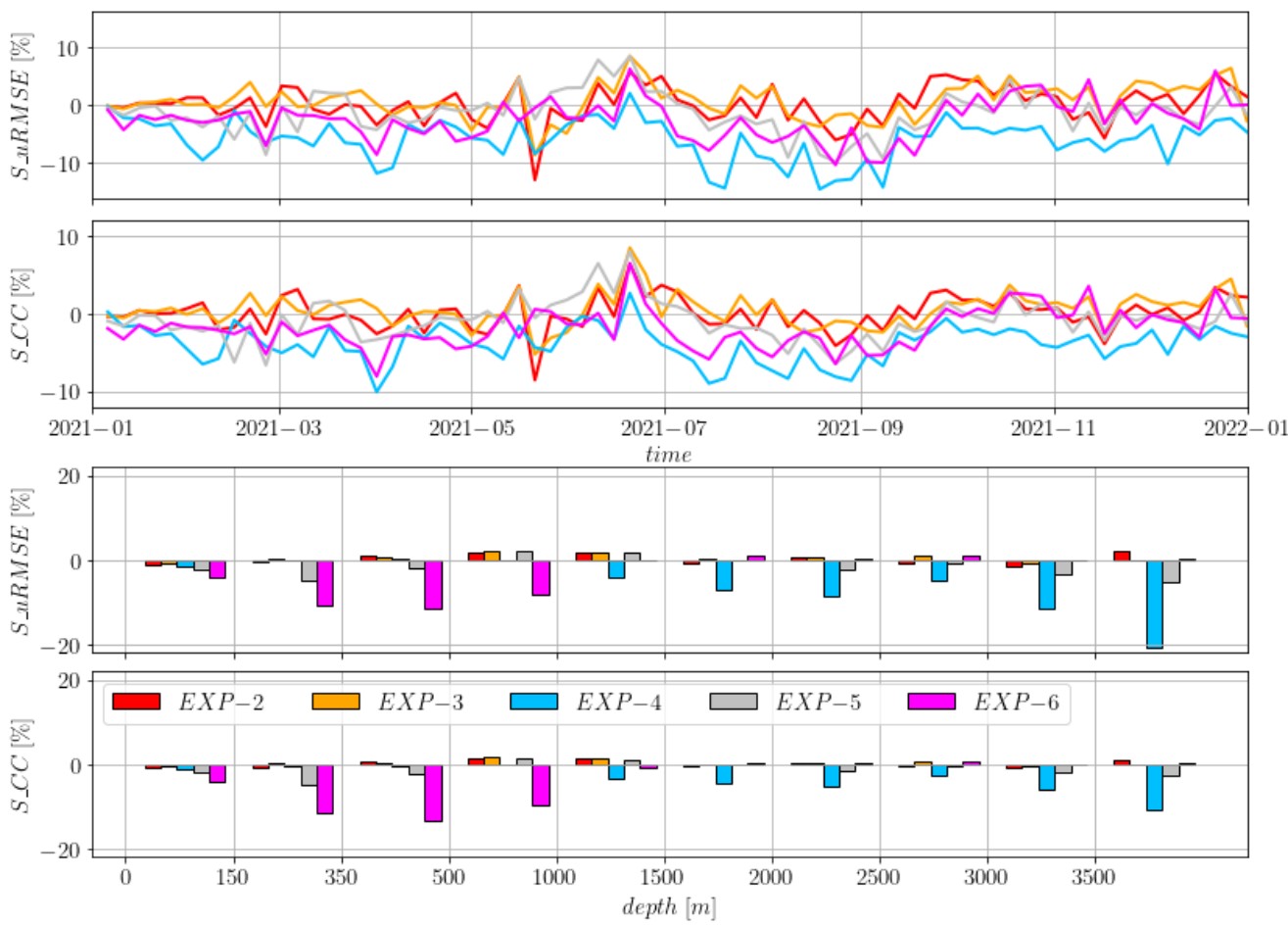

**Fig.8** *S_uRMSE* and *S_CC* indices for % of improvement of the SLA errors for the different experiments w.r.t Exp-1. Top two panels: time-series. Bottom panels: average indices for bathymetry classes.





**Fig.9** Second set of experiments SLA time series error statistics. Top panel: unbiased root-mean-square error. Middle panel: correlation coefficient. Bottom panel: standard deviation error. Colour code is provided in the middle panel legend.



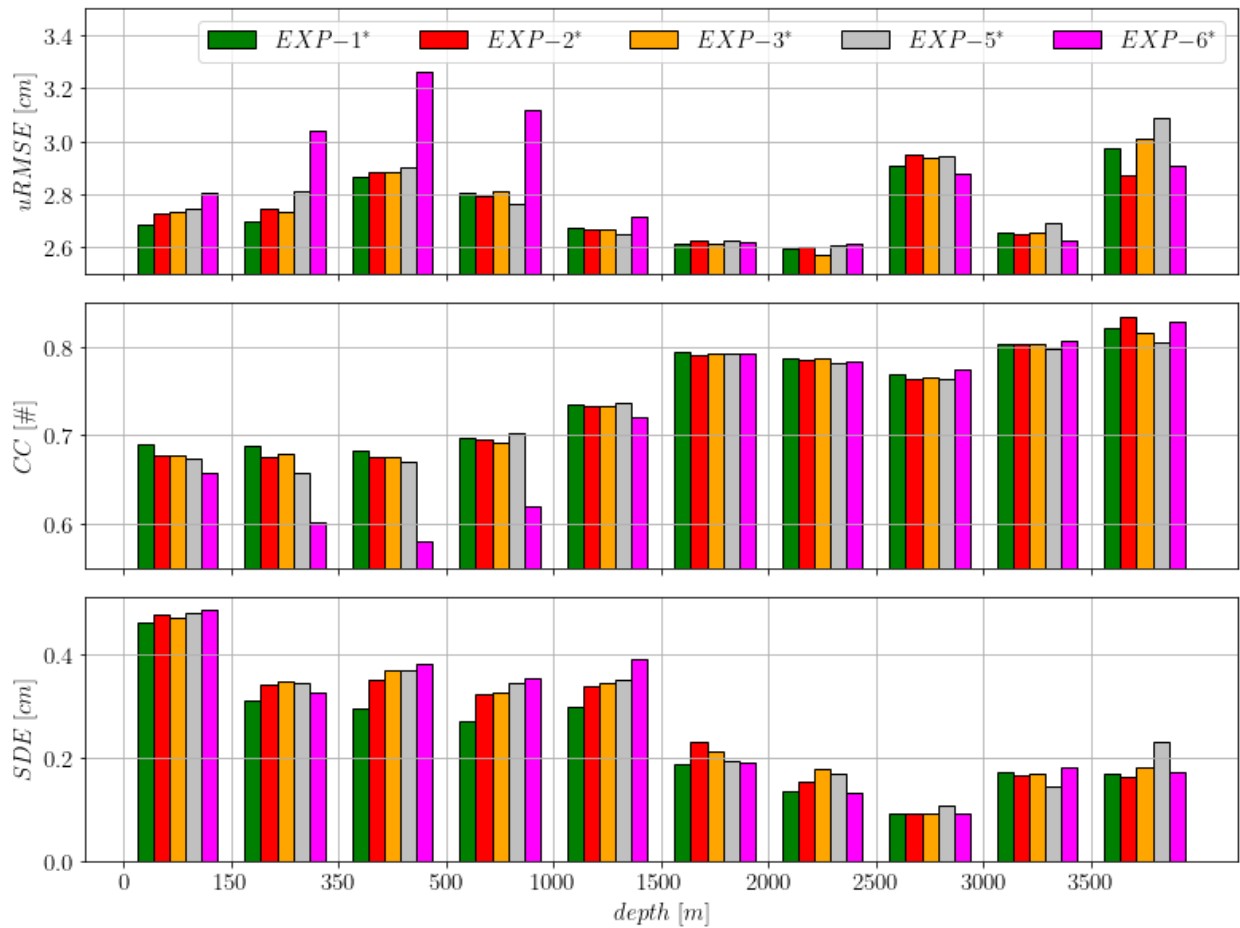

**Fig.10** SLA statistics as a function of the ocean depth. Top panel: unbiased root-mean-square error. Middle panel: correlation

coefficient. Bottom panel: standard deviation error. Colour code is provided in the top panel legend.






**Fig.11** Probability density function (Y axis) of the number of iteration (X axis) needed for the OceanVar2 minimization algorithm to converge. For each experiment, the lighter and darker shaded areas correspond to the 90% and 50% of the events respectively. The vertical lines are the medians of each distribution. Exp-1 to Exp-6 from top to bottom.







**Fig.12** CPU time per core as function of number of cores for Exp-2 and Exp-4. Solid lines represent the average CPU time, while shaded areas indicate the maximum and minimum CPU time for each number of cores.



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
