# Peer review of "Implementation and evaluation of sea level operators in OceanVar2.0: an open-source oceanographic three-dimensional variational data assimilation system"

_EGUsphere, 2025_

## Author Comment (AC2)

We thank the Editor for the clear guidance on the revision of the manuscript. We will provide detailed answers to the referees' comments separately.

**Additional Evidence for Code Correctness**

As requested, to provide evidence of the code's correct implementation, we have performed a series of idealized experiments analysing misfits and residuals. Instead of a full twin experiment, we designed specific tests for the assimilation of SLA tracks and Temperature/Salinity profiles. We specifically analysed the sensitivity of the residuals to the prescribed observational error. For these tests, synthetic observations ($y$) were generated at the analysis time ($t_a$), allowing for a direct assessment.

With $x_b$ as the model background and $\delta x$ as the increments, the analysis ($x_a$) is defined as:

$$x_a = x_b + \delta x$$

The misfits ($\boldsymbol{d}$) and residuals ($\boldsymbol{r}$) are then:

$$\boldsymbol{d} = \boldsymbol{y} - \boldsymbol{H}[\boldsymbol{x_b}]$$

$$\boldsymbol{r} = \boldsymbol{y} - \boldsymbol{H}[\boldsymbol{x_a}]$$

Consistent with our realistic experiments, the observation operator ($\boldsymbol{H}$) is a bilinear interpolator. We have chosen not to include these idealized experiments in the main manuscript as we believe they are not central to its narrative. However, we agree they provide a crucial check of the code's functionality and are happy to share them here as supplementary proof of correctness. We believe these additional analyses address the Editor's concerns and provide sufficient evidence of the correctness and robustness of the code used in our manuscript.

**Test 1: SLA track Assimilation**

To specifically verify the SLA operators, we assimilated a synthetic SLA track representing the data after bias and tidal signal removal. We performed the minimization using both the barotropic model and the dynamic height operators and using 3 different observational error values: 3 cm (as in the submitted manuscript); 5 cm and 8 cm.

Figure 1 shows the results. The top panel shows that the analysis residuals (coloured lines) are consistently smaller than the initial misfit (red line) and fall below the corresponding prescribed observational error (dashed lines). The central panel visualizes the background, the synthetic truth, and the resulting analyses. In the bottom panel the differences between the residuals ($\boldsymbol{r}|_{DH} - \boldsymbol{r}|_{BM}$) confirm that the difference in SLA residuals produced by the two operators is minimal. This is expected, as the primary impact of the choice of operator is on

the projected temperature and salinity profiles, as shown in Figure 3 of the submitted manuscript.

[Figure]

***Figure 1****: SLA Assimilation Test. **Top panel**: Initial misfit (red) and analysis residuals (solid-coloured lines) for experiments with varying observational errors (dashed lines). **Middle panel**: Synthetic "true" observation (red), the model background (magenta), and the resulting analyses (coloured lines). **Bottom panel**: Difference between residuals obtained using the barotropic and dynamic height operators. The numbers in the legends indicate the observational error used in the specific experiment (m).*

**Test 2: Temperature and Salinity Profile Assimilation**

While the manuscript's focus is not on in-situ data, we conducted similar tests for Temperature (*T*) and Salinity (*S*) profiles. Assimilating a single *T* or *S* observation is trivial, but assimilating a vertical profile is more complex because our system (OceanVar) enforces multivariate vertical correlations via pre-computed Empirical Orthogonal Functions (EOFs) in the Background Error Covariance (BEC). To create a clear test, we generated the synthetic observations to yield misfits that are consistent with the vertical structure of the BEC. In sensitivity experiments, we assimilated these profiles with prescribed observational errors of 0.2, 0.4, and 0.6 °C for temperature, and corresponding errors of 0.02, 0.04, and 0.06 psu for salinity.

The results in Figure 2 confirm the code functions correctly. For both temperature (left) and salinity (right), the analysis residuals decrease as the prescribed observational error is reduced, consistently remaining below the error threshold.

[Figure]

**Figure 2:** *Temperature and Salinity Profile Assimilation Test.* **Top Panels:** *Initial misfit (red) and analysis residuals (coloured lines) for Temperature (left) and Salinity (right) for different prescribed observational errors.* **Bottom Panels**: *Synthetic "true" profiles (red), model background (magenta), and the resulting analysis profiles (coloured lines).*

---

## Author Comment (AC3)

R: The study describes the development of OceanVar2.0, that is an oceanographic data assimilation method provided with open source software written for highly parallel computing environment. The manuscript further compares the method's performance in the Mediterranean Sea with respect to different options for assimilating Sea Level Anomaly (SLA) observations. The performance of assimilating SLA observations is particularly important in the Mediterranean Sea and many other regional seas and the global ocean, because often they represent the largest observational data set available in real time. I think that the study is well written. It provides an important insight into the improvements obtained by sophisticated application of a variational data assimilation scheme in oceanography that can be used as a reference for future applications of OceanVar2.0 and development and testing of other oceanographic data assimilation schemes. In particular, it highlights how applying a dynamical barotropic ocean model to simulate SLA perturbations within the model of the background error covariance matrix may significantly improve the variational data assimilation performance with respect to more commonly used simpler assumptions. I recommend the publication of the manuscript in Geoscientific Model Development after addressing a few minor comments.

*A: We thank the Anonymous Referee #1 for their careful reading of our manuscript and for their constructive comments. We greatly appreciate their positive assessment and are pleased that they found our work to be a useful contribution to the scientific community.*
*We have carefully considered all of their suggestions and have made the necessary changes to improve the manuscript, as detailed in the point-by-point responses below.*

R: Minor comments:

1. R: Line 140 and several other lines: I guess that DB08 should be DP08.
   *A: Thanks for pointing out this typo. The manuscript will be corrected accordingly.*

2. R: Line 270: Can removing the bias due to tides in observations and model forecasts increase the observational error? The two models have different bathymetries and use different computational methods for simulating the impact of tides. Is the atmospheric pressure forcing removed from SLA observations with the barotropic model?
   *A: We thank the reviewer for this important comment. We agree that the accuracy of the observational error is a critical component of any successful data assimilation scheme. We did not modify the SLA observational error to account for the removal of tides for few key reasons. Tides in the Mediterranean Sea are minimal and constitute only a small fraction of the overall SLA signal. Second, our method avoids assimilating SLA data in regions shallower than 100m. This exclusion is particularly important because it prevents the assimilation of data from the few shallow areas where tidal signals might be amplified. As shown in Figure 2 of the submitted manuscript, we provide an example of SLA with and without tides. The bottom panel of this figure demonstrates the model's ability to reproduce tides. Along-track tidal gradients are well reproduced in the model simulation, and the difference between the modelled and observed tidal signal is nearly constant. By removing the mean residual along each satellite track, this bias is effectively eliminated and should be very similar to the misfit computed without tides, which is why we decided not to modify the observational error when including tides. Regarding the atmospheric pressure, we followed the same procedure outlined in Dobricic et al. (2012). Since our numerical model is already forced by atmospheric pressure, we do not remove this effect from the SLA observations. Also in this case, the mean residual is removed along each satellite track in the assimilation scheme. This process effectively removes large-scale oscillations, including those from atmospheric*

*pressure, as well as the unknown steric height signal. We have also clarified that the barotropic operator in OceanVar2.0 remains forced only by the vertically integrated buoyancy force resulting from temperature and salinity variations.*

*We believe this setup is physically consistent and produced satisfactory results in our experiments. However, we completely agree with the reviewer that SLA observational error is a crucial factor for the assimilation scheme's success. This is why we have designed the software to provide great flexibility in prescribing this parameter. To improve clarity for the reader, we have revised the manuscript to better explain the observational error, the procedures we used, and the rationale behind our choices. We will add the following sentence in 4.2 Section. "The results presented in Fig. 2 demonstrate that our model accurately reproduces along-track SLA tidal gradients, with the difference between the modelled and observed tidal signal being nearly constant. We effectively remove this along-track bias by subtracting the mean residual along each satellite track. This critical step ensures that the resulting misfit is very similar to the one computed without tides. Based on these considerations, we decided not to modify the SLA observational error, maintaining consistency with previous work (Escudier et al., 2021). Consequently, all the SLA data have an associated error of 3 cm regardless of the satellite and the geographic distribution."*

3. R: Lines 380-390: Problems with vertical stratification might be also due to the calculation of vertical EOFs with difficulties to provide correct temperature and salinity increments from SLA assimilation during summer. Are there alternatives for using EOFs calculated from long model simulations in shallow areas?

*A: We thank the reviewer for this insightful comment. We agree that the use of Empirical Orthogonal Functions (EOFs) to model the vertical component of the Background Error Covariance (BEC) matrix can be a limitation, particularly in shallow areas and during the summer, when vertical stratification changes rapidly. The current version of OceanVar2.0 does not yet offer a built-in alternative to EOFs for representing the vertical error covariance. This is why, as a mitigation step, we decided to exclude regions shallower than 100m from the assimilation. While we recognize this remains a limitation of the current system, the modular design of OceanVar2.0 allows for the easy replacement of the EOFs with other, more sophisticated methods. Addressing this constraint is one of our highest priorities for future development, especially for coastal regions. We have added a brief sentence discussing this limitation and our future plans directly where the EOFs are introduced in the manuscript. "However, the persistence of this error maximum suggests a limitation in the current formulation of the Background Error Covariance. Specifically, the static, climatological nature of the EOFs used to model the vertical error component struggles to fully capture the rapidly evolving stratification and strong vertical gradients characteristic of the summer mixed layer. Future development will prioritize replacing these EOFs with more dynamic and stratification-aware operators to address this deficiency."*

4. R: Line 500: Does the minimizer converge more slowly, because the barotropic model is slightly non-linear or because it includes more complex dynamical processes requiring a slower convergence?

   *A: We thank the referee for this interesting and subtle question regarding the convergence performance. We agree that the observed tendency, though numerically small, is a point worth noting and discussing. Since this figure also drew commentary from another referee, we have slightly restructured the entire discussion section to first acknowledge the finding that the difference in the median values is small when compared to the overall daily variability. However, we believe the physical mechanism behind this consistent, minimal increase is relevant. As we do not have an analytical or definitive answer, we have maintained a cautious, speculative explanation in the revised text. We agree that this observation is worth mentioning, and the text now includes a possible explanation for this small, consistent difference: "The observed increase in the median number of iterations for the barotropic model schemes is likely due to the barotropic model's inherent physical complexity, which results in a more intricate optimization landscape for the minimizer to navigate."*

5. R: Lines 503-510: Is the land-see mask used to optimise the domain decomposition? According to Fig. 1, with a higher granularity many subdomains may become completely over land and this can reduce the scalability of the software. Are some subdomains computationally more demanding than the others, for example, due to extra computations near the coast? Eventually, the evaluation of the parallel performance could show the scalability of different parts of the software (e.g. barotropic model, geostrophic adjustment etc.).

   *A: We appreciate the detailed questions regarding our domain decomposition and the computational load across subdomains. The land-sea mask was not used to optimize the domain decomposition in these experiments. The decomposition was based purely on the model grid geometry, and we recognize that optimizing based on the land-sea mask is a valuable suggestion for future development to improve load balancing. Our analysis confirms that subdomains completely covered by land still required execution time comparable to the other domains. Regarding the computational load across subdomains, we found that there were no significant differences in computational demands based on geometric peculiarities, such as proximity to the coast. Instead, the computational load of a subdomain, and thus the total execution time, was primarily determined by the overall number of assimilated observations. This finding is consistent with the global nature of the variational solver and the number of iterations required for convergence. To provide a more complete assessment of scalability, we have added a new panel to Figure 12 showing the scalability of the most computationally significant routines. This analysis reveals that the limited overall speedup is not due to the domain decomposition being over land but is entirely due to the diffusive filter routine. Addressing this computational bottleneck is a priority for the future releases.*

---

## Author Comment (AC4)

1. R: The manuscript describes the application of sea level anomaly assimilation utilizing an incremental 3D-Var assimilation scheme. The manuscript introduces the common incremental 3D-Var scheme and the modeling of the background error covariance matrix, which utilizes a series of covariance operators. A particular aspect are two variants of the covariance operator for sea level, one uses dynamic height, while the other uses a barotropic model. Also the observation error covariance matrix is described. The main focus of the manuscript are the numerical experiments applying the data assimilation scheme to assimilate sea level anomaly data into a model for the Mediterranean Sea which includes the simulation of tides. Here, aspects like correcting assimilation innovations for tides and the sensitivity with regard to the configuration of the covariance operator for sea level anomaly, e.g. regarding the used reference or rejection depth levels are assessed. Finally, also the variation of the number of iterations in the optimization with the chosen configuration of the covariance operator and the parallel compute performance are discussed. The manuscript concludes that only the sea level operator using the barotropic model is sufficiently able to provide good assimilation results in the model domain due to its strongly varying bathymetry, while the dynamic height operator has limitations due to the higher sensitivity to the required specification of the level of no motion. The study also finds that the using barotropic model only leads to very small time increases.

   *A: We thank the Anonymous Referee #2 for carefully reading our manuscript and providing a comprehensive and constructive summary of our work. We greatly appreciate their positive assessment of the importance of this study. We believe that incorporating the detailed feedback provided has significantly improved the clarity, rigor, and overall quality of the revised manuscript. Our detailed responses, addressing every comment and critique, are provided below.*

2. R: The manuscript is overall well structured and the experiments are carefully done. Unfortunately, my experience with sea level operators for variational data assimilation is too limited to give a statement whether the application of a barotropic model operator is new. Actually, the authors refer to 'DB08' for this operator, but DB08 is undefined. I suppose that they mean the work by Dobricic and Pinardi from the year 2008 (thus rather 'DP08'), which introduces an operator using a barotropic model. DP08 did not include the code, but given that this work was published about 18 year ago by some of the original developers of OceanVar this at least gives the impression that the methodology is not new.

   *A: We apologize for the typo and thank the anonymous referee for the correction. The referee's interpretation is correct: the reference should be to (Dobricic and Pinardi, 2008). We have corrected this reference throughout the manuscript. The referee is correct that the barotropic model operator was first introduced by DP08. We have now rephrased several parts of the manuscript to stress that the novelty of our contribution lies not in the creation of the individual operator, but in the engineering and architectural unification of the OceanVar architecture. Previously, different development branches of OceanVar implemented and maintained these operators independently, often resulting in configuration inconsistencies and preventing a truly direct comparison of their performance. In OceanVar2.0, we have assembled all these individual developments into a single, cohesive codebase. This restructuring allows us for the first time to present a direct and rigorous comparison between the barotropic*

*model operator and the "classical" dynamic height approach. We believe this unique side-by-side analysis of two methods is a significant contribution to the field.*

3. R: The major weakness of the manuscript is that it claims to present the open source software OceanVar2.0 including an assessment of the sensitivity of sea level assimilation on the covariance operator for sea level. In fact, the manuscript clearly fails to present the software. The link to the software repository is provided and there are statements that it has a modular structure and, for example, produces for quality control of observations. However, any details on the structure of OceanVar2 as a software are missing. Further, the variational assimilation algorithm is mathematically described, but without hints on its actual implementation. Only section 6 "Performance and Parallelization" is more computationally oriented, but represents only a small part of the manuscript and includes very little detail. To this end, the actual topic of the manuscript is the effect of the sea level covariance operators while the software is just a side aspect. Thus, the manuscript is less a 'Development and technical paper' but rather an application paper.

*A: We thank the Anonymous Referee #2 for this candid and valuable critique regarding the manuscript's overall focus. We acknowledge that the initial manuscript presentation, driven by our enthusiasm for finally uniting previous developments into the single, open-source codebase, overemphasized the software release aspect. To address this critical issue and align the manuscript with its central scientific objective— a decision made following direct guidance from the Editor— we have strategically rewritten the manuscript. We will adopt the new title: "Implementation and evaluation of sea level operators in OceanVar2.0: an open-source oceanographic three-dimensional variational data assimilation system" We have revised the abstract, introduction, and relevant sections to clarify that the goal is not to present OceanVar2.0 as a new piece of software, but to leverage its new unified structure to conduct the first-ever direct comparison of the barotropic model operator and the dynamic height operator. This unique comparative analysis, available for the first time on an identical codebase, is the central contribution. The focus is now squarely on the comparative results and performance, thereby fully addressing the referee's concern.*

4. R: My recommendation on the manuscript is that the authors should decide whether they like to publish this work, in a slightly revised form, as a study on the effect of the covariance operators (if this is sufficiently novel) or like to revise it into a manuscript onto the actual software. The first variant would involve less work (a minor revision), while the second variant would involve a major revision to include sufficient details on the code structure and functionality in combination with a significant shortening of the experimental part into a demonstration of an application. Both variants should be suitable for GMD and it would be the authors' decision in which direction to go. (In the review system, I will mark this as a 'major revision', but the revision for the first variant would only be minor.)

*A: We thank the referee for the careful critique regarding the focus and structure of the manuscript. We acknowledge that the initial manuscript structure created confusion about its intended classification, leading to the impression that it was an incomplete model description paper. We have since engaged in a detailed discussion with the Editor regarding the journal's expectations. As per the Editor's guidance, the manuscript will remain categorized as a Development Paper, as this type is appropriate*

*for the main content. However, to fully address the referee's concern, we are revising the title, abstract, and introductory elements to shift the focus squarely onto the new developments implemented in OceanVar2.0 —specifically the code unification and the comparative analysis enabled by these new capabilities—rather than attempting a full description of the model as a whole. To satisfy the referee's request for more context on the software without making the manuscript overly dense with implementation details, we are incorporating additional descriptions of the newly implemented features directly into the relevant sections. Furthermore, we confirm that the additional analyses required by the Editor to better support the correctness and robustness of the code have already been completed and submitted as reply to the Editor's comment.*

R: Major comments (my comments are mainly aimed for the first revision variant - the application of studying the effect of the covariance operators).

5. R: Overall, the authors should show more care when preparing the manuscript. This is a manuscript with 13 authors, and the . With so many authors, an error like writing 'DB08' instead of 'DP08' should not happen if only one of the authors would have read the manuscript carefully. There are also other places, where the manuscript does not leave the impression of a careful proofreading. Generally, all co-authors are responsible for the quality of a submission and should take this responsibility seriously.

   *A: We sincerely apologize for the error (DB08 instead of DP08) and the resulting impression of a lack of care in the manuscript's preparation. We fully accept the referee's critique regarding the shared responsibility of all co-authors for the quality and accuracy of the submission. We have since conducted a complete and thorough internal review and proofreading of the entire revised manuscript to eliminate this and any similar editorial oversight. We confirm that the error has been corrected throughout the text, and we are confident that the revised version adheres to the highest standards of presentation and accuracy.*

6. *R: On OceanVar2.0 as an assimilation 'scheme': The manuscript denotes OceanVar2.0 as a 'variational ocean data assimilation scheme'. I don't think that this is correct. Rather, OceanVar2.0 is the software that implements an incremental 3D-Var scheme. Thus, the actual scheme is incremental 3D-Var. From the description in the text, this incremental 3D-Var also seems to be a standard implementation as, e.g., described in the review by Bannister (2017). Also the use of covariance operators is common practice. To this end, it is the actual implementation which is particular, but the algorithm itself is not new. Please rephrase the manuscript accordingly.*

   *A: We thank the referee for making this important technical distinction. We agree that OceanVar2.0 is the software that implements the incremental 3D-Var scheme. The actual algorithm is indeed the standard incremental 3D-Var, which is not novel. We have conducted a systematic review and correction throughout the entire manuscript to ensure that we correctly distinguish between the two. We now refer to OceanVar2.0 as the "variational data assimilation system" or "software" that implements the "incremental 3D-Var scheme," thereby clarifying that the novelty lies in the implementation, unification, and analysis capabilities of the software.*

7. R: Section 2: The section reads irritatingly because it discusses a common incremental 3D-Var scheme with FGAT, which is widely used in ocean data assimilation, but the text is written as if a new method is introduced. It would be useful to clarify the typo of the mathod at the very beginning of the sections. To improve the structure, I recommend to move the explanation of FGAT (mentioned in line 111 with some details in lines 115-118) to the end of the section. Please also improve the explanation and clarify the notation of the equation in line 117. I don't see how this does actually explain FGAT (the equation uses innovations in an interval, while FGAT means that each innovation is computed at the time of the observation.)

A: We thank the referee for the detailed guidance on Section 2's structure and clarity. We acknowledge that the original text inadvertently suggested we were introducing a new method rather than describing the widely used standard incremental 3D-Var scheme with FGAT. We now explicitly state the use of the standard incremental 3D-Var scheme in the introduction of the section. Following the suggestion, we moved the detailed explanation of FGAT to the end of the section for improved structural flow, and we completely rewrote the FGAT explanation while clarifying the mathematical notation of the equation in line 117. We confirm that the revised explanation accurately reflects that each innovation is computed at the time of the observation. We believe these revisions significantly improve the precision, structure, and readability of the section.

8. R: Sections 3 and 3.1: As mentioned before, the text refers to DB08, which probably means Dobricic and Pinardi (2008), which is defined as 'DP08' in line 50, for the covariance operators, but then not used. The sections include the sea level operator using a linear barotropic model (line 181). Here it would be important to clarify how far this is new. It appears contradictory that the previous developers of OceanVar published the use of the barotropic model, but apparently did not include it in OceanVar.

A: We apologize for the typo that has been corrected in the revised version of the manuscript. We also thank the referee for requesting this important clarification on the history of the sea level operator, as this is an important distinction that was not clear in the original text. While the developers of OceanVar (DP08) performed both the theoretical work and the initial coding for the barotropic sea level operator, the original implementation was not carried forward or maintained in subsequent mainline versions of the code. The operator was, in effect, absent from recent production versions of the system. Our work in OceanVar2 focused on the re-introduction, integration, and parallelization of this operator to function efficiently within the new framework. Specifically, the barotropic model code was ported and optimized, maintaining its general mathematical structure from the DP08 publication. Its re-introduction as a fully integrated and optimized option is a key novelty of this version, enabling the rigorous comparative analysis presented here. We have revised the text in Section 3.1 to reflect this precise development history.

9. R: Section 5 (Here, I focus on the overall content, while more specific comments are further below):

The discussion of the results is quite detailed, but parts of it are also only short and add little insight. I recommend to overall shorten this part and to focus on the relevant insights, skipping more superficial discussions. In particular:

- For me it is unclear why the authors decided to perform two sets of experiments, one without correcting velocities and one with such correction. Actually, multivariate data assimilation is the usual standard today. Thus, updating the velocity components when assimilating sea level data appears to be common practice. Given that no particular insights are found from the separate discussion of the two sets (the order of the skill is not significantly different), I recommend to focus on one set. This should preferably be the multivariate case updating also velocities.

*A: We thank the referee for the prompt to sharpen our discussion and focus on the relevant insights. While we agree to streamline the discussion, we believe the comparison of both experiment sets (Set 1 and Set 2) is essential because it illustrates a critical operational design choice in multivariate assimilation. Both experiment sets are carried out with the same multivariate framework, and the distinction lies in which variables receive a direct analysis increment. In Set 2, we apply the full multivariate increment to all state variables (T, S, eta, u, and v). While this maximizes the direct observational impact, it amplifies the system's sensitivity to the balance terms approximated within the Background Error Covariance matrix, leading to the numerical challenges we observed. In Set 1, we intentionally constrain the analysis to apply direct increments only to the mass-field variables (T, S, and eta). The velocities (u and v) are still corrected indirectly because the model's physics subsequently adjusts the momentum field through the forecast cycle. By preventing direct increments to the velocities, we effectively damp potential contamination or instability arising from an imperfect B matrix approximation. Thus, our comparison illustrates the essential operational trade-off: stability and robustness (Set 1) versus maximal direct observation impact (Set 2). We will clarify this precise technical rationale in the revised manuscript to underscore the importance of this comparison.*

10. R: - The relative improvement score for the correlation coefficient, S_CC, appears to result in unreasonably large values. This is due to the division by the reference correlation, which in cases where the correlation in the free run is below 0.1, leads to high values. Given that correlations have a well-defined upper limit of 1, the normalization by the reference correlation is not required. It would adequate to simply assess the difference in the correlations between the free run and each assimilation experiment, and I recommend to do so. The results would be mainly the same, but without weirdly looking improvements of up to nearly 10000 % for a bounded quantity like the correlation coefficient. Perhaps, one would also see the differences of the effects in different assimilation experiments in Figure 7 more clearly.

*A: We thank the referee for identifying this critical issue in the calculation and interpretation of the relative improvement score for the correlation coefficient. We agree that normalizing a bounded quantity like correlation by a near-zero reference correlation leads to unreasonably large and misleading percentage values. To correct this, we have adopted the recommended approach. This change resolves the issue of the exaggerated percentages, provides a more meaningful metric for the improvement of a bounded quantity, and clarifies the visual differences between the assimilation experiments in Figure 7. Figure 7 and the corresponding text have been updated*

*accordingly. We confirm that Figure 8 was not affected by this issue and remains unchanged.*

11. R: - Figure 8 shows the scores relative to the baseline assimilation experiment Exp-1. The figure as well as the related discussion is mainly redundant with the information included in Figure 7 and its related discussion. Differences between the different assimilation experiments are already visible in Fig. 7. It is just that Fig. 8 quantifies these differences directly. However, this also seems to amplify apparent differences, which are actually very small in Fig. 7. To this end, I recommend to omit Fig. 8 and the related discussion. If the authors like to point out particular differences that are currently only discussed in relation to Fig. 8, they can include these into the discussion relating to Fig. 7.

    *A: We thank the referee for the careful evaluation and the suggestion regarding the potential redundancy between Figure 7 and Figure 8. While we agree that the underlying information is present in both figures, we respectfully disagree with the recommendation to omit Figure 8, as we believe it is crucial for robustly defending our conclusions. Figure 7 primarily demonstrates the large overall improvement of all assimilation runs compared to the free-run; this high baseline performance visually compresses the subtle but important distinctions among the assimilation experiments. Figure 8, by showing the scores relative to the baseline assimilation run (Exp-1), is specifically designed to quantify and highlight the impact of the changes introduced in subsequent assimilative experiments. The referee notes that this relative view "amplifies" apparent differences, but we contend that this quantification is necessary and objective to distinguish scientifically significant variations that are visually minimized in Figure 7. Therefore, to provide the clearest insight into the performance hierarchy among our assimilation experiments, we prefer to retain Figure 8 and its related discussion.*

12. R: - Table 2 gives an overview of scores, but the text only refers to the table without ever discussing details. I don't see that the table contains relevant information beyond what was already discussed in relation to the figure. Hence the table can be dropped without loosing relevant information.

    *A: We thank the referee for this suggestion. We agree that Table 2 primarily summarized information already presented and discussed in detail through Figures 7 and 8. To streamline the results section, we have removed Table 2 from the manuscript.*

13. R: - Figures 9 and 10: As mentioned before, I don't see an added benefit of discussing the two different sets of experiments (with and without updating the velocities). In fact, the discussion of the detailed Figures 9 and 10 are only 12 lines of text. There does not seem to be additional insight except that the "differences between the experiments are less pronounced" (lines 474/475). This insight could also be included into the manuscript as a single sentence without adding figures.

    *A: Thank you for this constructive comment. We understand your concern regarding the length of the discussion for Figures 9 and 10. However, we believe that the inclusion of Set 2 (with velocity corrections) and its comparison to Set 1 is essential for validating our system's operational viability and for a complete scientific comparison. The key justification for retaining the comparison lies in two critical findings. First, the*

*immediate numerical instability of Dynamic Height at 150m with velocity correction is a major finding that directly addresses the operational challenge of implementing full multivariate data assimilation. This result validates the cautious approach taken in Set 1 (mass-field-only correction), proving that the stability of a forecasting system is not guaranteed when introducing direct momentum increments, even with a robust background error covariance matrix. We believe, this instability alone justifies the inclusion of the two sets as it defines a crucial operational boundary. Furthermore, the comparison quantifies the actual benefit of direct velocity correction over the more stable indirect correction. We show that all stable experiments in Set 2 outperform their counterparts in Set 1 and that the poorest-performing setup benefits the most from the direct velocity correction in deep water. This demonstrates that the increments can sometimes compensate for other methodological weaknesses. We agree that the previous discussion was too concise to highlight these critical insights. We have expanded and clarified the discussion of Figures 9 and 10 in the manuscript to emphasize the instability of Exp-4, the overall error reduction, and the specific comparative performance across bathymetric classes. We believe the inclusion of these figures and the expanded discussion is necessary to fully address the operational relevance of the implementation.*

14. R: - Section 5 contains particularly many grammatical errors. It would be good if the authors would spend some time to improve the writing quality of this section.

   *A: We sincerely apologize for the grammatical errors in Section 5 and the resulting impact on readability. We fully accept this critique. To address this, we did not just perform a cursory check; we have conducted a comprehensive, dedicated review and revision of the entire manuscript, with particular attention paid to Section 5. We are confident that the revised manuscript, especially this crucial results section, now presents a high standard of clarity and grammatical quality.*

   R: Section 6: This section discusses the compute performance. It should be of not particular relevance if the focus of the paper is targeted to the application of SLA assimilation. Apart from this, there are some critical points:

15. - Line 485 states that "Rigorous testing has been conducted to guarantee bit-for-bit (BFB) reproducibility ...", but then in lines 487/488 it is clarified that the global matrix-vector multiplication used in the solver algorithm precludes such reproducibility. To this end, it is unclear for what parts the reproducibility was ensured. For simple distribution of arrays, the reproducibility is always ensured, while global parallel sums (and global matrix-vector multiplications include such) do not ensure reproducibility. Please clarify this aspect.

   *A: Thank you for requesting this clarification. The referee is correct that global parallel sums inherent to the minimizer preclude BFB reproducibility. We clarify that BFB reproducibility was ensured for all components of the codebase except the cost function minimizer. The minimizer is the only part of the system that requires global parallel communication (a matrix-vector multiplication). The system offers a compile-time option that allows the user to execute the L-BFGS minimizer serially while the rest of the system remains fully parallelized via MPI. This serial execution aggregates variables from all domains and forces a fixed order of floating-point operations, thereby ensuring BFB reproducibility for testing. This feature was implemented*

*specifically to allow users to verify that all other parallelized components of the data assimilation system (e.g., array distribution, observation operators, covariance modeling) are implemented robustly and reproducibly. We have revised the text in the manuscript to clearly state this distinction.*

16. R: - Figure 11 and related text: Figure 11 shows distributions of the number of required iteration of the solver for the different experiments. This information is far more detailed than the related text. The main insight from the figure is that the number of iterations can vary strongly between about 12 and 45. However, the median number of iterations varies only between 24 and 26 iterations, which is insignificant given the wide spread. To this end, I don't see any point in the discussion in lines 495-502. The insight is instead that the different configurations do not lead to any relevant difference in the required number of iterations.

    *A: Thank you for your sharp assessment of Figure 11 and the related discussion. We agree that our initial text focused disproportionately on the small differences in the median values, misrepresenting the figure's primary insight. The referee correctly identifies that the wide distribution of daily required iterations (spanning 12 to 45) means that the difference between the median convergence speeds (which vary only from 24 to 26) is indeed insignificant. We have fully revised the sentences to reflect this finding. The revised text now emphasizes that the convergence performance is robustly consistent across all tested configurations. We clarify that these small differences in the median values appear insignificant when contrasted with the wide day-to-day variability, which is driven by fluctuations in data availability and the associated misfit values over the year-long testing period.*

17. R: - Figure 12 and related text: The text claims that "Up to 36 cores, the experiment with the dynamic height operator consistently outperformed the one using the barotropic model" (lines 507/508). However, the figure shows that the difference of the average run time is nearly identical with a difference that seems to be below 1 second for most cases. At the same time, the run times show a large variation of 10 seconds or more. Thus, there does not seem to be any statistically significant difference in the average run time. The authors also discuss that the performance improves up to 36 or 72 cores depending the on the experiment, but shows not further improvement for larger numbers of cores. This discussion omits the important fact that there are cases where the run time increases for more than 36 cores. In addition, the figure shows that the speedup is in fact very limited. For the blue line of Exp-4 only a speedup of about 8 is obtained when increasing the number of cores from 1 to 36 (from about ~40 seconds to ~5 seconds). This implies a parallel efficiency of about 22%. Thus, the scalability of the algorithm is very limited despite the large model grid of 1307x380x141 points. I recommend that the authors also discuss the limited scalability and its possible reasons. Also the discussion of the variation of run times would be more relevant to discuss than the insignificant difference of the average run times.

    *A; We thank the referee for their comprehensive analysis of Figure 12 and their candid assessment of our initial discussion. We agree that the previous text misrepresented the findings by overstating the minimal average run-time difference and omitting a necessary discussion of the limited parallel efficiency. We have revised lines 503–510 to correct these points:*

- *Run Time Difference: We now state that the difference between the dynamic height and barotropic model run times is minimal even if consistent.*
- *Limited Scalability: We explicitly discuss the limited scalability. The observed speedup for Exp-4 is approximately 8x for 36 cores, corresponding to a parallel efficiency of about 22%. We also highlight the performance plateau reached around 36 cores, where run time deteriorates at higher core counts due to increased communication overhead.*

*To address the limited scalability and the observed wide run-time variability (the shaded area in the top panel), we have introduced a new panel to Figure 12 that details the performance of individual routines. This analysis demonstrates that the bulk of the OceanVar2 code scales nearly optimally as speed-up is nearly linear. The overall speedup is, moreover, entirely constrained by the diffusive filter routine. The bottleneck originates because the first implementation of the diffusive filter operator presented in this manuscript employs a direct solver (LU) within a Dimensional Splitting approach. This choice was made to simplify implementation. Crucially, this routine's limited scaling and extreme sensitivity to the geometry of the domain decomposition is the primary cause of the wide run-time variability shown in the top panel of Figure 12. We have revised the text to highlight this critical operational dependency.*

R: Section 7: As for previous recommendations, I recommend that the authors carefully reconsider the scope of the manuscript.

18. - Please also avoid contradictory statements like that "Computationally the barotropic model is more expensive than the dynamic height operator" (line 533) followed by "large time-step significantly limiting the computational demand." (line 535). This statement says that the barotropic model is more costly, but the higher cost is very small. One could write directly in a concise form that the barotropic model does only lead to a very small increase in run time.

*A: Thank you for pointing out the contradictory nature of lines 533 and 535. We agree that the statements are redundant and confusingly phrased. Our intention was to convey that while the barotropic model is structurally more complex than the dynamic height operator, its net increase in computational cost is minimal. This is specifically due to the stability of the semi-implicit scheme, which permits a large time step. We have revised the text to state concisely that although the barotropic model is computationally more complex, its impact on the total run time is minor because the large time step significantly limits the demand. This clarification has been included in the revised manuscript.*

19. R: - In line 539 it is stated "The OceanVar2 code is stable, robust, its previous versions have been largely documented in several scientific papers...". If the manuscript is about OceanVar2, which is a revised code, it should be irrelevant that previous publications already described previous versions of OceanVar. This holds even more as no previous papers are referred to (This keeps me from checking if it is true that previous version of OceanVar core are actually documented and accessible.) Please ensure that the statement is consistent and focused.

*A: Thank you for seeking clarification on this statement. We agree that claiming the stability of the new OceanVar2 version by referencing older versions without context or citation was logically incomplete. Our intention was to establish the long-term, peer-reviewed heritage of the code, not simply assert that its predecessor was stable. The OceanVar system is not a new creation but an evolution built upon a robust, well-documented foundation. We have revised the text to clearly state that the mathematical and algorithmic core of the system, first presented in Dobricic and Pinardi (2008), has been widely validated and used in operational frameworks. The revised sentence now focuses on the confirmed stability and robustness of the current version, while providing verifiable references for its heritage.*

20. R: - The last sentence of the section shortly points to future developments. This sentence is speculative 'Future developments could explore', but also superficial. I recommend to drop it or to replace it with aspects that are more concrete.

    *A: Agree, the sentence is removed in the new version of the manuscript.*

    R: Further comments:

21. lines 32/33: The authors cite De Mey and Robinson (1987) on the requirement of 'advanced extrapolation algorithms' for the 'effective integration' of satellite altimetry and sea surface temperature data. I have the impression that this statement and citation is outdated. Data assimilation advanced strongly since the year 1987. Nowadays, the assimilation of altimetry and surface temperature is common practice. I don't see any point in stating that nearly 40 years ago, this was considered a challenge.

    *A: Agree, text modified accordingly.*

22. R: lines 44/45: the text states "...in developing shared software tools like PDAF (Nerger et al., 2005), ROMS-4DVAR (Moore et al., 2011), DART ..." Here, ROMS-4DVAR does not fit into the list. While PDAF and DART are general DA frameworks, ROMS-4DVAR is a particular implementation for the model ROMS and not otherwise usable. Please revise this list. Further, the reference for PDAF is incomplete and one cannot find this proceedings paper based on this incomplete reference. It can also be useful to cite Nerger and Hiller (2013), which is a peer reviewed publication.

    *A: Agree, text modified accordingly.*

23. R: lines 46/47: Following the mention of some DA software tools mentioned above, the text states "However, the increasing complexity of data assimilation problems, particularly with the adoption of ML, demands continued research and development of new tools." In this sentence the authors seem to claim that the systems mentioned before do not include 'continued research' and that they cannot adopt ML. This is clearly incorrect for PDAF and DART, which are regularly updated with new releases. Please rephrase this to avoid such claims.

    *A: Agree, text modified accordingly: "The data assimilation community has made substantial strides in developing widely used software tools and specific implementations. General frameworks like PDAF (Nerger and Hiller, 2013) and DART (Data Assimilation Research Testbed, Anderson et al., 2009) have contributed*

*significantly to the field. Additionally, a notable model-specific implementation like ROMS-4DVAR (Moore et al., 2011) has been instrumental in advancing ocean data assimilation. Each of these tools offers unique capabilities, and the increasing complexity of data assimilation problems—particularly with the adoption of ML—drives the continuous development and innovation of both existing and new tools in the community, as evidenced by recent research (Martin et al., 2025)."*

24. R: line 54: The reference Coppini et al. (2023) is not a proper citation. Is this a preprint; is there a published version?

    *A: Thanks for noticing. Reference updated in the new version of the manuscript.*

25. R: line 56: The statement on OceanVar "has been used to test and develop numerous new schemes and features" is not useful in this form and seems to be mainly aimed at adding references in which the authors are involved. I recommend to either include more information on the 'new schemes and features' or rather to remove this statement.

    *A: We agree; in the revise version of the manuscript we added the needed information.*

26. R: lines 63/64: The authors described line 60 that OceanVar2 "involved comprehensive testing and debugging" and here state "This rigorous development positions OceanVar2.0 as what we consider a leading advancement in ocean data assimilation". Apparently, the authors consider testing and debugging to be particular. However, this is just common practice in software development. I get the impression, that the statement might imply that OceanVar was previously not carefully tested and debugged, thus the development did not follow common development practices. This potential weakness however, does not give a basis to claim that the testing and debugging is a "leading advancement in ocean data assimilation". Thus, I recommend to remove this statement.

    *A: We agree the sentence has been entirely rewritten and in the new sentence we omitted "we consider a leading advancement in ocean data assimilation".*

27. R: line 75: 'revisited' should likely be 'revised'

    *A: Thanks. However the sentence has been reviewed in the new version of the manuscript.*

28. R: lines 77/78: It is state that "OceanVar2 is applied to a newly developed Mediterranean Sea circulation forecast model". This statement looks misleading. The model does not seem to be a 'new' model, but an advancement of the previous model system resulting from adding tidal forcing.

    *A: We agree with the referee. We changed the sentence from "Mediterranean Sea circulation forecast model" to "Mediterranean Sea forecasting System".*

29. R: line 82: 'Section 1' should be 'Section 2'

    *A: Thanks. We correctly the manuscript accordingly.*

30. R: line 90: It's stated: "x is the true state". I don't think that this is correct. 'x' is the state what is to be estimated by the DA process. The true state is unknown and one would not be able to change it.

   *A: Thanks. We correctly the manuscript accordingly.*

31. R: line 93: It's odd to see the definition of particular state vector variables in a general introduction to the 3DVar method. I recommend to more this to an appropriate location describing the experimental setup.

   *A: We agree with the referee. We moved the definition of the state vector in Section 4.*

32. R: line 112: "The minimum of J(\delta x) in (3) is obtained for x_a = x." Please take into account that there is no 'x' in Eq. (3).

   *A: Thank for pointing out this inconsistence. We correctly the manuscript accordingly.*

33. R: line 121: 'defined' should likely be 'definite'

   *A: We thank the anonymous referee. The text has been modified accordingly.*

34. R: line 191: The sentence in this line is an outlook, which doesn't fit into the scope of this section. Perhaps, it can be mentioned in the conclusions.

   *A: We thank the anonymous referee. The text has been modified accordingly*

35. R: line 222: Here, Brankart et al. (2009) is cited with regard to the observation error, which looks misleading. Obviously, this study did not introduce the concept of observation errors, or the distinction of measurement and representation error, which is used for much longer time. Perhaps, one can cite Brankart et al. (2009) with 'see' in the sense of a review. Alternatively, the authors can look for a more original paper.

   *A: We thank the anonymous referee. Brankart et al 2009 has been replaced with a more fundamental work: Cohn, S. E., 1997: An introduction to estimation theory. J. Meteor. Soc. Japan, 75, 257–288*

36. R: lines 234-239: Here statements like "Several options are implemented in the OceanVar2 to shape the observation error and the interested reader is asked to consult with the code manual, available with the code." or the description of possible quality control options are very superficial. If the authors decide to focus the manuscript on the application, then these descriptions should be revised to describe particular methods that are used in the study. If the authors decide to revise to present the software OceanVar2, more concrete details would be needed.

   *A: We thank the referee for identifying the superficiality of this section. We have revised the text to address this issue. We removed the invitation to consult the code manual and rewrote the section to focus on the specific quality control (QC) and data pre-processing methods employed in the experiments presented in this work.*

37. R: line 256: The sentence in this line looks redundant to that in line 250. I recommend to remove it.

*A: Thank; in the revised version of the manuscript we removed the sentence.*

38. R: Figure 1/line 259: The figure shows the satellite track for SLA data over 21 days. Showing this long period seems to contradict the fact that daily assimilation is done. It would be more meaningful to show the satellite tracks for one day. This would give an indication of the data coverage.

*A: We agree with the referee that illustrating the data coverage for a single, representative daily assimilation cycle is much more meaningful than showing a longer period. We have revised Figure 1 to address this point. We retained the visualization of the total available in-situ data and the 21-day altimetry tracks as contextual information. We have added a snapshot of the actual daily data coverage for 11 August 2021 to illustrate the data density during a single assimilation run. The figure caption has been updated to clearly distinguish between the overall data pool and the daily operational coverage.*

39. R: Lines 267/268: The manuscript states "This paper offers a solution to the assimilation of satellite altimetry in a model with tides, showing that a filtering procedure can be accurate enough..." I'm wondering why it should be useful to filter out tides using some tidal components? Since tides are both in the observations and the model wouldn't it be better to actually assimilate the full signal, hence correcting also the tides? Please comment on why the filtering is your preferred approach (I also recommend to write 'This work' rather than using the colloquial 'This paper')

*Thank you for the excellent question regarding the filtering of tidal signals. We agree that assimilating the full signal is the ultimate goal; however, in a standard 3D-Var scheme like OceanVar2, this is dynamically challenging and often detrimental. The core reason for filtering the tidal signal is physical and algorithmic. The Background Error Covariance matrix is designed to project Sea Level Anomaly (SLA) corrections onto the slow baroclinic components. If the SLA misfit contains a tidal signal, this error is primarily due to external gravity waves whose magnitude is sensitive to static model parameters (like model bathymetry and bottom friction). Applying the standard matrix to correct a tidal error would therefore cause the tidal energy to be incorrectly projected onto the baroclinic fields, resulting in spurious, unphysical temperature and salinity increments. Therefore, filtering the high-frequency tidal signal is the necessary approach to maintain the physical integrity of the baroclinic analysis and ensure the stability of the model increments. We have clarified this rationale in the revised manuscript text.*

40. R: Line 288 and elsewhere: The authors use 'simulation' to denote an experiment without assimilation. This terminology is irritating given that all experiments are actually simulations. I recommend to use a more common and less irritating terminology like 'free' or 'free run' to denote the experiment without data assimilation.

*A: Text corrected accordingly, thank the anonymous referee.*

41. R: Line 300: Please be more specific about how the observation errors were tuned. Simply referrring to Desroziers et al. (2005) looks insufficient.

*A: Thank you for requesting more specificity on the tuning process. We agree that simply citing the method is insufficient. We have revised the manuscript text to detail the procedure used: We applied the Desroziers' diagnostic method (Desroziers et al., 2005) in an iterative manner. We began with observation errors from a previous system, then ran repeated cycles of the assimilation system. We used the calculated innovations and residuals from those runs to apply the Desroziers' formula, and this process was iterated until the estimated observation errors converged. This yielded the final, robust, and monthly-varying observation error matrix R.*

42. R: Line 303: Observation errors of 0.1degC and 0.02psu appear very small given that there should be significant representation errors given the model resolution of about 4 km. Please comment on the reansing for the chosen values.

*A: Thank you for your comment regarding the small magnitude of the deep observational errors. We acknowledge that these values appear small, but they are not arbitrarily chosen. They are the result of the iterative tuning process using the Desroziers' diagnostic method that we described in the revised manuscript text for the previous comment. We believe that the additional information now included in the manuscript regarding the iterative Desroziers' method provides sufficient justification for these error values, and therefore, we do not propose further modifications to the stated observational error values in the manuscript text.*

43. R: Lines 304-322: The structure of the text in which the experiments are introduced from line 304 onwards is irritating. It is mentioned that two 'sets of experiments' are performed, but then six experiments are described. Only later, in line 319, one learns that the six experiments are actually 'one set' and that there is a second set. I recommend to already define the difference of the two sets after the initial sentence in line 304 and to clarify that each 'set' consists of six actual experiments.

*We thank the referee for pointing out the confusing structure in the introduction of the experiments. We agree that the delayed definition of the two sets caused irritation. We have restructured the section, clarifying immediately after the introductory sentence that two sets, each consisting of six individual experiments, were performed, and we define the difference between these two sets upfront to improve clarity and flow.*

44. R: Line 330: The first sentence is very difficult to read due to sub-clauses that break the logical flow. Please reformulate it for clarity.

*A: Text corrected accordingly, thank the anonymous referee.*

45. R: Lines 345-361: I'm wondering why the explanation of the scores is included within the description of the actual results. It should be more meaningful to include it as a subsection in Sec. 4.

*A: We agree with the anonymous referee. In the new version of the manuscript, we added Section 4.3 "Performance metrics" and moved the description of the scores there.*

46. R: Lines 368/369 "A slight but consistent improvement is noted in the CC." What makes the improvement 'slight'? Changing a correlation from 0.3 to 0.7 looks more than 'slight'. Please avoid such subjective classifications and concentrate on the numbers.

   *A: Text corrected accordingly, thank the anonymous referee.*

47. R: Line 370: "The SDE in the simulation is generally negative". This statement contradicts the figure, which shows positive and negative values of the SDE. It just that the amplitude of negative SDE is higher than those of positive SDE. Please correct the statement.

   *A: Text corrected accordingly, thank the anonymous referee.*

48. R: Line 373: "The SLA ... statistics were clustered according to ocean depth". This statement is misleading. Actually, to my understanding from later text, the clustering is not for depth intervals, but for regions in which the ocean floor is at a certain depth (line 472 states 'clustered according to the bathymetry'). Please clarify this here.

   *A: Text corrected accordingly, thank the anonymous referee.*

49. R: Line 375; "more constant" - this is one of the examples showing little care in writing. 'constant' cannot be raised.

   *A: Text corrected accordingly, thank the anonymous referee.*

50. R: Line 386: Here, "In Fig. 6 the..." repeats the beginning of the previous sentence. Further, it is unclear why a new paragraph is started here.

   *A: Text corrected accordingly, thank the anonymous referee.*

51. R: Line 425: "These results highlight the challenges associated with choosing an appropriate level of no motion". I don't see where the "challenges" are. Basically the experiments show which level does perform well. Unfortunately, the experiments do not seem to provide insight into why a certain level performs better. An obvious aspect is that the omission of observations due to the specified level counters possible positive effects of including observations with a possibly sub-optimal level. This aspect finds little attention in the discussion and should be made more explicit.

   *A: Thank you for seeking further clarification on the "challenges associated with choosing an appropriate level of no motion." We recognize that the previous text was too general and did not sufficiently explain the underlying physical constraints. Our experiments are specifically designed to highlight the fundamental theoretical limitation of the Dynamic Height Operator in complex, confined domains. This operator imposes a critical methodological compromise. For instance, maintaining a conservative, deep level of no motion guarantees the physical consistency but forces the mandatory exclusion of a large fraction of valuable SLA observations. Conversely, adopting a shallower level of no motion maximizes observation coverage, yet the resulting analysis is compromised due to the violation of the theoretical assumptions. The differing performance across experiments provides empirical evidence of the consequences of violating these physical constraints. The superior stability and*

*performance of the Barotropic Model Operator confirms its methodological advantage, as it successfully overcomes this level of no motion constraint by assimilating all available SLA data regardless of depth. We have revised the manuscript to explicitly detail this methodological compromise and its physical consequences.*

52. R: Line 441: It is stated "... see that the Exp-4 and Exp-5 produce, in general, worse results, and the worsening is amplified as the depth increases and the level of no motion decreases." I have the impression that this is what we simply expect fromphysics: Assuming an unrealistic level of no motion yields worse results. I'm wondering that the authors just set the level of no motion to some essentially arbitrary value, while it should be known at which approximate depth the actual level of no motion exists. Including more physical insight might help here.

    *A: Thank you for this excellent point. We certainly agree that a reliable operational system must adhere to known physical constraints. We assure the referee that the choices for the level of no motion in Exp-5 (350m) and Exp-6 (1000m) were based on strong physical and operational precedents (Rio et al., 2014; Coppini et al., 2023). The rationale behind Exp-5 (Level of No Motion at 150m) was not arbitrary, but purely experimental: to rigorously test the sensitivity and stability limits of the data assimilation system when attempting to maximize Sea Level Anomaly (SLA) data inclusion in shallow areas. The physical insight gained from the poor results of Exp-4 is precisely the confirmation we sought: violating the core physical assumption of a realistic level of no motion significantly degrades the quality of the solution, and, as shown by the instability of Exp-4\* in Set 2, introduces critical numerical instability when coupled with direct momentum corrections. The experiment thus serves to establish the minimum physically required depth for the level of no motion in our specific assimilation setup, thus defining the operational boundary between robust performance and numerical failure. We have clarified the text describing the setup for Exp-4 to explicitly state this experimental rationale.*

53. R: Lines 466/467: "the relative performance of the different experiments seems to be confirmed". Given that the actual values are available there should be no need to write a speculative statement like 'seems to be confirmed'. The authors can clearly assess the performance and provide a concise statement.

    *A: Text corrected accordingly, thank the anonymous referee.*

54. R: Line 481: The text states that "To optimize computational performance ...adopts a domain-decomposition". Actually, domain-decomposition does not "optimize" computation performance, but it can improve it. Please rephrase.

    *A: Text corrected accordingly, thank the anonymous referee.*

55. R: Line 484: Please add a reference for MPI

    *A: Reference added.*

56. R: Line 556: 'OP' should likely be 'PO' for Paolo Oddo.

    *A: Text corrected accordingly, thank the anonymous referee.*

57. R: I recommend to also carefully check proofread the manuscript. There are various places where singular or plural are incorrectly used, or where the grammar is incorrect. While there are such errors throughout the manuscript, Section 5 including its sub-sections contains particularly many grammatically errors like incorrect word ordering.

*A: We thank the referee for this suggestion. The revised version of the manuscript has been checked carefully.*

---

## Author Response (AR2)

Editor Comment:
Thanks for all your hard work on reworking the paper.

Showing that the code works as intended is actually an essential feature of a GMD paper!! See the Manuscript types webpage for evidence. So, a write-up on the extra work that you completed to answer my editor's comment could most definitely have been included in the manuscript.

However, since you have already done so much revision to improve the manuscript, in this case I will be satisfied if you can modify the information you provided in the interactive review so that it fits into the general context, and upload it as a supplement. You must also include a paragraph in the main manuscript that summarises the issue and references the supplement.

Reply:

Thank you very much for your feedback and for the guidance throughout the revision process. I appreciate the clarification regarding the expectations for GMD papers and fully understand the importance of demonstrating that the code functions as intended.

In response to your comment, I have revised the material provided during the interactive review so that it fits within the overall context of the manuscript, and I have uploaded it as supplementary material. I have also added a paragraph to the main manuscript that summarises the issue, describes the additional work carried out to address it, and provides a clear reference to the supplement.

Specifically, in Section 4.2, immediately after the description of the sensitivity experiments analysed in the manuscript, we added the following sentence:

"To complement the sensitivity experiments described above, we performed a set of idealised tests with synthetic observations to verify that the assimilation system behaves as expected under controlled conditions. In these tests, the consistency of the implementation is assessed by analysing how the analysis residuals vary in response to the prescribed observational error. Full details are provided in the Supplement."